# The AP-1 transcription factor Fosl-2 drives cardiac fibrosis and arrhythmias under immunofibrotic conditions

Mara Stellato[1], Matthias Dewenter[2], Michal Rudnik[1], Amela Hukara[1], Çagla Özsoy [3,4], Florian Renoux[1], Elena Pachera[1], Felix Gantenbein[5], Petra Seebeck[5], Siim Uhtjaerv[1], Elena Osto[6], Daniel Razansky[3,4], Karin Klingel[7], Joerg Henes[8], Oliver Distler[1], Przemysław Błyszczuk[1,9] & Gabriela Kania [1✉]

Fibrotic changes in the myocardium and cardiac arrhythmias represent fatal complications in systemic sclerosis (SSc), however the underlying mechanisms remain elusive. Mice over-expressing transcription factor Fosl-2 (Fosl-2tg) represent animal model of SSc. Fosl-2tg mice showed interstitial cardiac fibrosis, disorganized connexin-43/40 in intercalated discs and deregulated expression of genes controlling conduction system, and developed higher heart rate (HR), prolonged QT intervals, arrhythmias with prevalence of premature ventricular contractions, ventricular tachycardias, II-degree atrio-ventricular blocks and reduced HR variability. Following stimulation with isoproterenol Fosl-2tg mice showed impaired HR response. In contrast to Fosl-2tg, immunodeficient Rag2−/−Fosl-2tg mice were protected from enhanced myocardial fibrosis and ECG abnormalities. Transcriptomics analysis demonstrated that Fosl-2-overexpression was responsible for profibrotic signature of cardiac fibroblasts, whereas inflammatory component in Fosl-2tg mice activated their fibrotic and arrhythmogenic phenotype. In human cardiac fibroblasts FOSL-2-overexpression enhanced myofibroblast signature under proinflammatory or profibrotic stimuli. These results demonstrate that under immunofibrotic conditions transcription factor Fosl-2 exaggerates myocardial fibrosis, arrhythmias and aberrant response to stress.

[1] Center of Experimental Rheumatology, Department of Rheumatology, University Hospital Zurich, University of Zurich, Zurich, Switzerland. [2] Institute of Experimental Cardiology, University Hospital, Heidelberg, Germany. [3] Institute for Biomedical Engineering and Institute of Pharmacology and Toxicology, Faculty of Medicine, University of Zurich, Zurich, Switzerland. [4] Institute for Biomedical Engineering, Department of Information Technology and Electrical Engineering, ETH Zurich, Zurich, Switzerland. [5] Zurich integrative Rodent Physiology, University of Zurich, Zurich, Switzerland. [6] Institute of Clinical Chemistry, University Hospital Zurich, Zurich, Switzerland. [7] Cardiopathology, Institute for Pathology and Neuropathology, University Hospital Tubingen, Tubingen, Germany. [8] Internal Medicine II, Division of Rheumatology, University Hospital Tubingen, Tubingen, Germany. [9] Department of Clinical Immunology, Jagiellonian University Medical College, Krakow, Poland. ✉email: gabriela.kania@uzh.ch

Systemic sclerosis (SSc) is a connective tissue disease characterized by dysregulation of the immune system, diffuse vascular lesions and fibrosis of the skin, but also of internal organs including lungs, oesophagus, kidneys, and the heart. Fibrosis is defined as the excessive accumulation of extracellular matrix (ECM) proteins, mostly collagen type I, expansion of ECM-producing stromal cells in the tissue and is associated with development of myofibroblasts characterized by high expression of alpha-smooth muscle actin (αSMA), periostin and ADAM12[1,2].

Cardiac involvement in SSc patients is often clinically asymptomatic; however, when it becomes clinically evident, it represents a poor prognostic factor and often leads to increased mortality, economic burden and depressing the quality of daily life[3,4]. The most frequent manifestations comprise inflammatory and fibrotic lesions involving all cardiac structures[5], including the conduction system. Abnormal electrocardiograms (ECGs) and arrhythmias are common in SSc patients[6], including atrioventricular (AV) blocks as the most frequent alterations[6]. In addition, QT prolongation has been reported as one of the most frequent manifestation of heart involvement in SSc patients[7].

However, the mechanisms leading to alteration of the cardiac conduction system in SSc remain unclear, causing challenges for the development of specific and targeted therapies for the patients at risk. Increased interstitial fibrosis in the cardiac tissue might separate myocardial bundles and lead to asynchronous propagation of the action potential resulting in conduction abnormalities and arrhythmias[8]. However, the prevalence of cardiac fibrosis and conduction system defects in SSc is challenging to be determined due to limitations of the applied diagnostic tools and diverse patient cohorts. In fact, previous studies have demonstrated that many electrophysiological cardiac alterations can be detected only by applying specific cardiac investigations like 24 h Holter monitoring[9].

Fos-related antigen 2 (Fosl-2, also termed Fra-2) belongs to the Fos protein family, which dimerize with factors of the JUN family, thereby forming the transcription factor complex Activator Protein 1 (AP-1). The AP-1 complex has been implicated in multiple biological processes, such as cell proliferation, differentiation and transformation[10]. Overexpression of Fosl-2 in mice induces fibrotic changes in the skin, lungs and the heart, and is recognized as an animal model of SSc[11–13]. A previous study described histological changes in the myocardium of Fosl-2tg, but cardiac function in terms of hemodynamic and cardiac electrical activity has not been investigated so far. Here, we aimed to characterize conduction abnormalities and arrhythmia susceptibility in Fosl-2tg mice and to elucidate the role of Fosl-2 and adaptive immune response in this cardiac pathology.

## Results

### Fosl-2tg mice develop myocardial fibrosis but show normal systolic and diastolic functions.
Fosl-2tg mice (developed by Sanofi) develop a systemic autoimmune phenotype with extensive leukocyte infiltrates in the skin, lungs, duodenum, and pancreas, and fibrosis in the thymus, salivary glands, and lungs, mostly in perivascular structures and to a lesser extent in the interstitium[14] (Suppl. Fig. 1). Here, we assessed the level of myocardial fibrosis comparing cardiac sections from 7- and 16–22-week-old Fosl-2wt and Fosl-2tg mice. Sirius red staining showed comparable collagen cardiac levels at week 7 and increased collagen deposition in Fosl-2tg hearts at week 16–22 (Fig. 1a and Suppl. Figs. 2a and 3a). To assess, if collagen deposition was accompanied by the activation of stromal cells, we analysed the expression of periostin and αSMA. Like for collagen, the expression of periostin and αSMA was comparable in the heart of 7-week-old mice, but significantly

increased in Fosl-2tg hearts at week 16–22 (Fig. 1a and Suppl. Figs. 2b, c and 3a). Analysis of cellular lysates confirmed higher total αSMA levels in Fosl-2tg hearts at week 16–22, but not at week 7 (Fig. 1b and Suppl. Fig. 3b).

We previously demonstrated that stromal (Ter119-CD31-CD45- = Lin-)/gp38+ cells represent type I collagen-producing fibroblasts in the mouse heart[15]. At week 7, the frequency of cardiac Lin-gp38+ fibroblasts was comparable in Fosl-2wt and Fosl-2tg mice (Suppl. Fig. 4). In all, 16–22-week-old Fosl-2tg hearts showed a higher number of Lin-gp38+ fibroblasts compared to Fosl-2wt hearts (Fig. 1d, e). Importantly, gp38+ cells correlated with collagen deposition and were localized in fibrotic regions of the Fosl-2tg heart (Fig. 1f, g). Furthermore, gp38+ fibroblasts in 16–22-week-old Fosl-2tg hearts showed increased levels and abundant co-expression with the myofibroblast markers αSMA, Adam12 and periostin (Fig. 1h). Fosl-2tg hearts displayed also higher expression of myofibroblast markers compared to Fosl-2wt hearts (Fig. 1h).

Fibrotic changes in hearts of Fosl-2tg mice were, however, not associated with cardiomyopathy or cardiac dysfunction. Heart weight/body weight (HW/BW) and heart weight/tibial length (HW/TL) measurements were comparable between Fosl-2wt and Fosl-2tg mice (Fig. 1c and Suppl. Fig. 3c). Echocardiography analyses showed no difference in systolic or diastolic functions between Fosl-2wt and Fosl-2tg mice, however Fosl-2tg mice revealed reduced LV volume and cardiac output (Suppl. Table 3).

### Fosl-2tg mice develop arrhythmias and rhythm defects.
Increased cardiac fibrosis may be associated with pro-arrhythmic changes[16]. Analyses of continuous recording of 24 h revealed that 16–22-week-old Fosl-2tg mice (with a gross phenotype) demonstrated a significantly higher heart rate (HR) and prolonged QT intervals compared to Fosl-2wt mice (Fig. 2a and Suppl. Figs. 5, 6). Moreover, Fosl-2tg mice showed significantly lower values of HR variability parameters describing the parasympathetic/sympathetic balance, i.e. NN intervals, standard deviation of all NN intervals (SDNN), the square root of the root mean square of the sum of all differences between successive NN intervals (RMSSD), NN6 and pNN6 parameters compared to Fosl-2wt mice (Fig. 2a). Importantly, Fosl-2tg mice developed cardiac arrhythmias with a higher prevalence of premature ventricular contractions (PVCs), ventricular tachycardias (VTs), and atrio-ventricular (AV) blocks second-degree (Fig. 2b–d). Peculiarly, QT intervals positively correlated with AV blocks, while QT intervals and AV blocks positively correlated with the disease phenotype and collagen deposition (Suppl. Fig. 7).

### Impaired response to stress of Fosl-2tg mice.
Based on radiotelemetry data and HR results under anaesthesia (Suppl. Table 3 and Movies 1–2), we suspected that Fosl-2tg mice might show altered response to stress. Indeed, Fosl-2tg mice showed a significantly lower HR increase compared to Fosl-2wt mice following stimulation with isoproterenol (ISO), which indicates an aberrant response to stress (Fig. 3a). Moreover, Fosl-2tg mice showed reduced response time after ISO stimulation (Fig. 3b, c and Suppl. Fig. 8a). Similarly, ex vivo Langendorff system measurement of HR in the isolated mouse hearts confirmed the abrogated response to ISO stimulation of Fosl-2tg hearts (Fig. 3d and Suppl. Fig. 8b).

### Alteration in the conduction system and gap junctions in Fosl-2tg mice.
Next, we performed the analyses of the gene expression in the cardiac conduction system. In Fosl-2tg hearts, we observed lower transcripts levels of genes involved in: the regulation of cardiac rhythmicity (Meis1, Nfia), the functional development, maturation, and homeostasis of the conduction system (Tbx3),

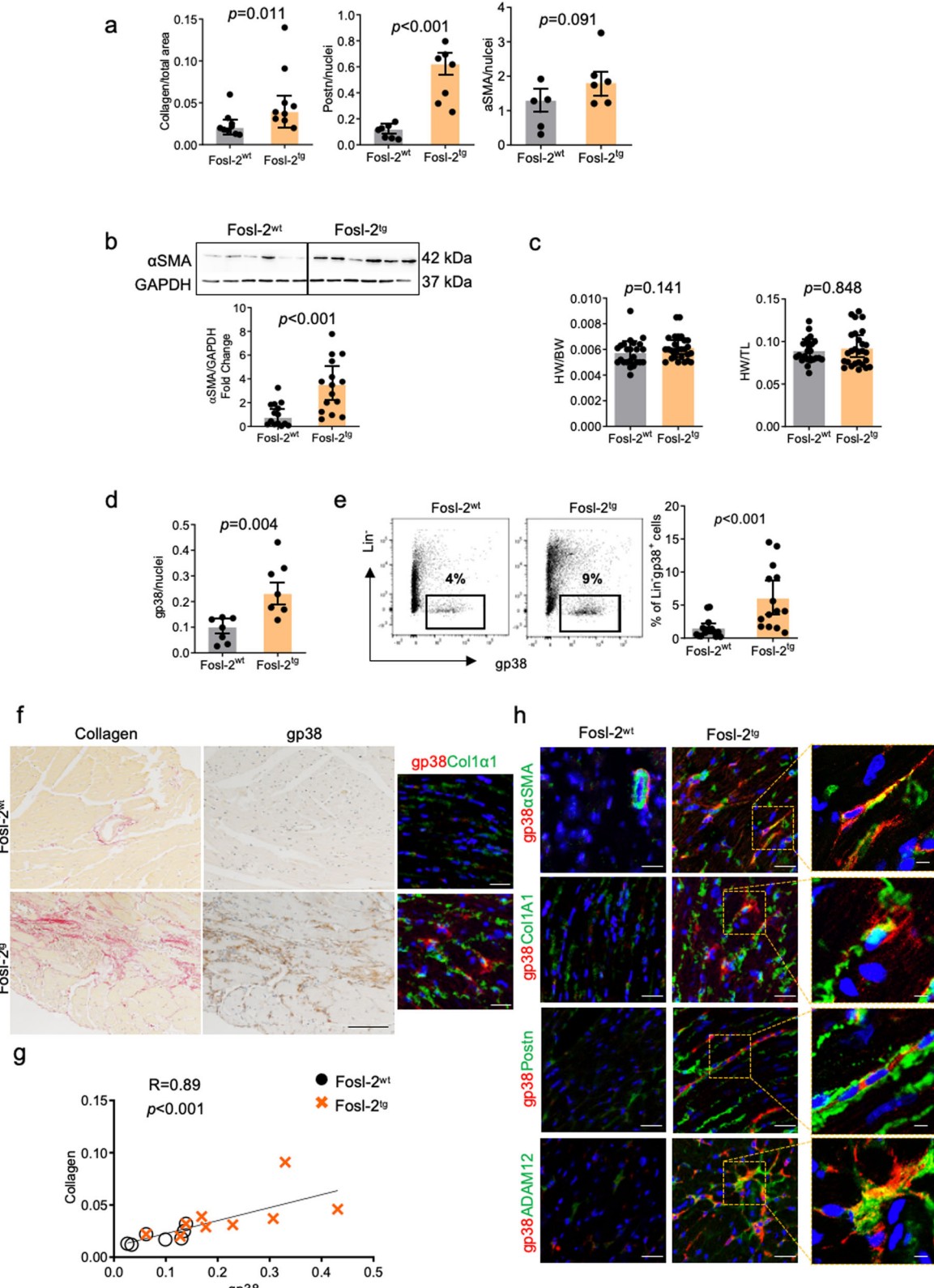

higher resting HR, atrial fibrillation and left ventricular mass (*Sox5*), long QT syndrome (*Scn10a*, *Kcnq1*), Brugada syndrome, sick sinus syndrome, and atrial fibrillation (*Scn10a, Nos1ap*) (Fig. 3e).

Furthermore, the immunofluorescence analysis revealed that in Fosl-2[tg] hearts contactin-2[+] Purkinje cells were surrounded by the deposited collagen (Fig. 4a, b). We hypothesized that the

expansion of cardiac fibroblasts might result in the observed defects of the conduction system causing a loss of cardiac ultrastructure and disruption of proteins composing gap junctions such as connexins (Cxs). In cardiac tissue of Fosl-2[wt] mice, Cx43 was localized in intercalated disks and co-localized with E-cadherin (E-cad) (Fig. 4c). In Fosl-2[tg] hearts, instead, the spreading of gp38[+] fibroblasts corresponded with the loss of

**Fig. 1 Lin⁻gp38⁺ cell phenotyping during SSc myocardial fibrogenesis. a** Quantification of Sirius Red staining for collagen, IHC for periostin (Postn) and αSMA of myocardial sections from 16 to 22-week-old mice ($n = 5$–9) (Mann–Whitney test, mean ± SEM). **b** Representative WB and densitometric analysis of αSMA protein content in homogenized hearts isolated from 16 to 22-week-old mice ($n = 14$–15, unpaired $t$-test, mean ± SEM). **c** Heart weight (HW)/body weight (BW) and HW/tibial length (TL) ratios at week 16–22 ($n = 24$–27, unpaired $t$-test, mean ± SEM). **d** Quantification of myocardial sections of gp38 IHC staining from 16–22-week-old mice ($n = 7$, Mann–Whitney test, mean ± SEM). **e** Representative flow cytometry pictures and corresponding quantification of Lin⁻gp38⁺ fibroblast frequency from the myocardium of 16–22-week-old mice ($n = 13$–14, Mann–Whitney test, mean ± SEM). **f** Co-localization of collagen and gp38 expression in myocardial sections ($n = 5$, scale bars: 100 μm) and IF of heart sections from 16 to 22-week-old Fosl-2$^{wt}$ and Fosl-2$^{tg}$ mice ($n = 5$, scale bars: 50 μm). **g** Spearman correlation between collagen expression quantified by Sirius Red staining and gp38 expression ($n = 7$–9). **h** Representative pictures of myocardial sections stained with the indicated antibodies ($n = 10$). In all IF staining, nuclei are stained with DAPI (blue), scale bars: 50 μm, insert scale bars: 10 μm.

Cx43 from the intercalated discs and its disorganized distribution (Fig. 4c). Noteworthy, total Cx43 protein levels did not differ between Fosl-2$^{tg}$ and Fosl-2$^{wt}$ hearts (Suppl. Fig. 9). Cx40 (Gja5) is a major gap-junction protein in the atrial myocardium and its altered expression is associated with increased propensity for arrhythmias. Fosl-2$^{tg}$ hearts showed abnormal expression and distribution of Gja5, mainly in the fibrotic regions with the presence of gp38$^+$ cardiac fibroblasts (Fig. 4d). In addition, we found that Fosl-2$^{tg}$ mice expressed more regulators of G protein signalling 4 (Rgs4) in the myocardium (Fig. 4e). These data may suggest that the expansion of cardiac fibroblasts in Fosl-2$^{tg}$ hearts is associated with the disruption of gap junctions and defects in the conduction system.

**Cardiac fibrosis and arrhythmia depend on autoimmune response in Fosl-2$^{tg}$ mice.** At the age of 12–16 weeks (females) and 16–22 weeks (males) Fosl-2$^{tg}$ mice develop systemic inflammation. At this time, Fosl-2$^{tg}$ hearts exerted mild inflammation demonstrated by higher numbers of CD45-expressing cells (Fig. 5a), statistically significant higher percentages of infiltrating Ly6G$^{int}$/Ly6C$^{int}$ monocytes and Lin⁻/CD11b$^+$ monocytes, lower percentages of CD11b⁻/MHCII$^{hi}$ dendritic cells (DCs) and CD45$^+$/Lin⁻ cells, and unchanged percentages of CD11b$^+$/MHCII$^{hi}$ DCs, Ly6G$^{hi}$ neutrophils, T cells and Ly6c$^{hi}$ monocytes (Fig. 5b; gating strategy in Suppl. Fig. 10). To assess the role of systemic inflammation in our model, Fosl-2$^{tg}$ mice were crossed with immunodeficient Rag-2$^{−/−}$ mice to generate Rag2$^{−/−}$Fosl-2$^{tg}$ mice (lacking T and B cells, but overexpressing Fosl-2). Rag2$^{−/−}$Fosl-2$^{tg}$ hearts neither showed increased collagen deposition nor expansion of activated periostin$^+$ or Lin⁻gp38$^+$ fibroblasts compared to Rag2$^{−/−}$Fosl-2$^{wt}$ hearts (Fig. 5c, d and Suppl. Fig. 11a). Similarly, HR and QT intervals, as well as HW/BW and HW/TL were not altered in Rag2$^{−/−}$Fosl-2$^{tg}$ (Fig. 5e and Suppl. Fig. 11b). As expected, Rag2$^{−/−}$Fosl-2$^{tg}$ hearts presented normal distribution of Cx43 in intercalated discs (Fig. 5f). These findings demonstrate that autoimmune response is a critical factor to induce cardiac phenotype in Fosl-2$^{tg}$ mice.

To dissect effects mediated by Fosl-2 overexpression and autoimmune response, we performed deep RNAseq of FACS-sorted cardiac fibroblasts from Rag-2$^{−/−}$Fosl-2$^{wt}$, Rag2$^{−/−}$Fosl-2$^{tg}$, and Fosl-2$^{tg}$ mice. To specifically address the role of systemic inflammation, we compared gene expression between cardiac fibroblasts obtained from Rag-2$^{−/−}$Fosl-2$^{tg}$ and Fosl-2$^{tg}$ mice. We detected 115 significantly deregulated genes (Fig. 6a). Gene ontology analysis of upregulated genes in Fosl-2$^{tg}$ (vs. Rag-2$^{−/−}$Fosl-2$^{tg}$) fibroblasts pointed to increased transcription factor pathways (Smad2,3,4, Rela, Myb, Irf8), activation of immunofibrotic pathways (IL-1-regulation of extracellular matrix; TNF-α-effects on cytokine activity, cell motility and apoptosis; TGF-β-regulation of extracellular matrix), and development of a pro-arrhythmogenic signature (Fig. 6a). In addition, the analysis of all deregulated genes pointed also to affected extracellular-matrix assembly, inflammatory response and chemokine-mediated cellular response (Supp. Fig. 12).

**Fosl-2 enhances the profibrotic signature in cardiac fibroblasts.** Next, we analysed genes deregulated by Fosl-2 overexpression in cardiac fibroblasts in the absence of inflammation. We detected 54 deregulated genes between Rag-2$^{−/−}$Fosl-2$^{wt}$ and Rag-2$^{−/−}$Fosl-2$^{tg}$ fibroblasts (Fig. 6b). Gene ontology analysis of upregulated genes in Rag-2$^{−/−}$Fosl-2$^{tg}$ (vs. Rag-2$^{−/−}$Fosl-2$^{wt}$) fibroblasts indicated increased transcription factor pathways (Smad2,3,4, Gata2, Mef2a, Clock), activation of profibrotic and rhythm-related pathways (Wnt interactions, tight-junction interactions, Bmal1-Clock activation circadian expression), and association with an SSc signature (Fig. 6b). The analysis of all deregulated genes pointed also to altered cytokine-mediated signalling (Supp. Fig. 12).

To evaluate the cellular phenotype and functions of cardiac fibroblasts, we analysed sorted Lin⁻/gp38$^+$ cells from hearts of 16–22-week-old mice and found slower proliferation and upregulated apoptosis (but unchanged cellular senescence) in Fosl-2$^{tg}$ cells (Fig. 6c–e). Furthermore, Fosl-2$^{tg}$ cardiac fibroblasts showed higher levels of αSMA total protein, αSMA-positive stress fibres, increased secreted collagen I and increased contraction capacity (Fig. 6f–i).

**GP38$^+$ fibroblasts are activated in hearts of SSc patients.** Endomyocardial biopsies (EMBs) from SSc patients (Suppl. Table 1) presented increased collagen deposition as compared to control EMBs from patients with healed myocarditis (Suppl. Table 2 and Fig. 7a). GP38 was barely detectable in control EMBs, whereas in SSc EMBs, GP38 expression was significantly increased (Fig. 7b), accompanied with increased expression of FOSL-2 (Fig. 7c). Of note, FOSL-2 was also expressed in the heart of SSc patients[11]. To assess the function of FOSL-2 in human cardiac fibroblasts, we used human foetal cardiac fibroblasts (fCFs). The majority of fCFs expressed GP38 (Fig. 7d) and following TGF-β stimulation upregulated FOSL-2 (Fig. 7e). Next, we overexpressed FOSL-2 in fCFs (Fig. 7f) but found unchanged αSMA protein levels (Fig. 7g). However, additional stimulation with TGF-β or LPS induced production of the myofibroblast marker αSMA in FOSL-2 overexpressing fCFs (Fig. 7h).

**Discussion**
Previous studies identified histological changes in the myocardium of Fosl-2$^{tg}$ mice, but not in other SSc mouse models, as sclerodermatous chronic graft-versus-host disease and Tight-Skin-1 mice[11]. Herein, we explored not only functional consequences of cardiac fibrosis in Fosl-2$^{tg}$ mice, but also defined the role of Fosl-2 in cardiac fibroblasts and pointed to a crucial role of systemic autoimmune response in the development of cardiac pathology. The relevance of FOSL-2 expression was further assessed in EMBs from SSc patients, demonstrating similarities between the cardiac phenotype in Fosl-2$^{tg}$ mice and SSc patients[11].

Our results highlighted increased prevalence of arrhythmic events in Fosl-2$^{tg}$ mice. Despite certain differences in electrophysiology between rodents and humans, essential mechanisms of stimuli generation and propagation are conserved[17]. Clinical

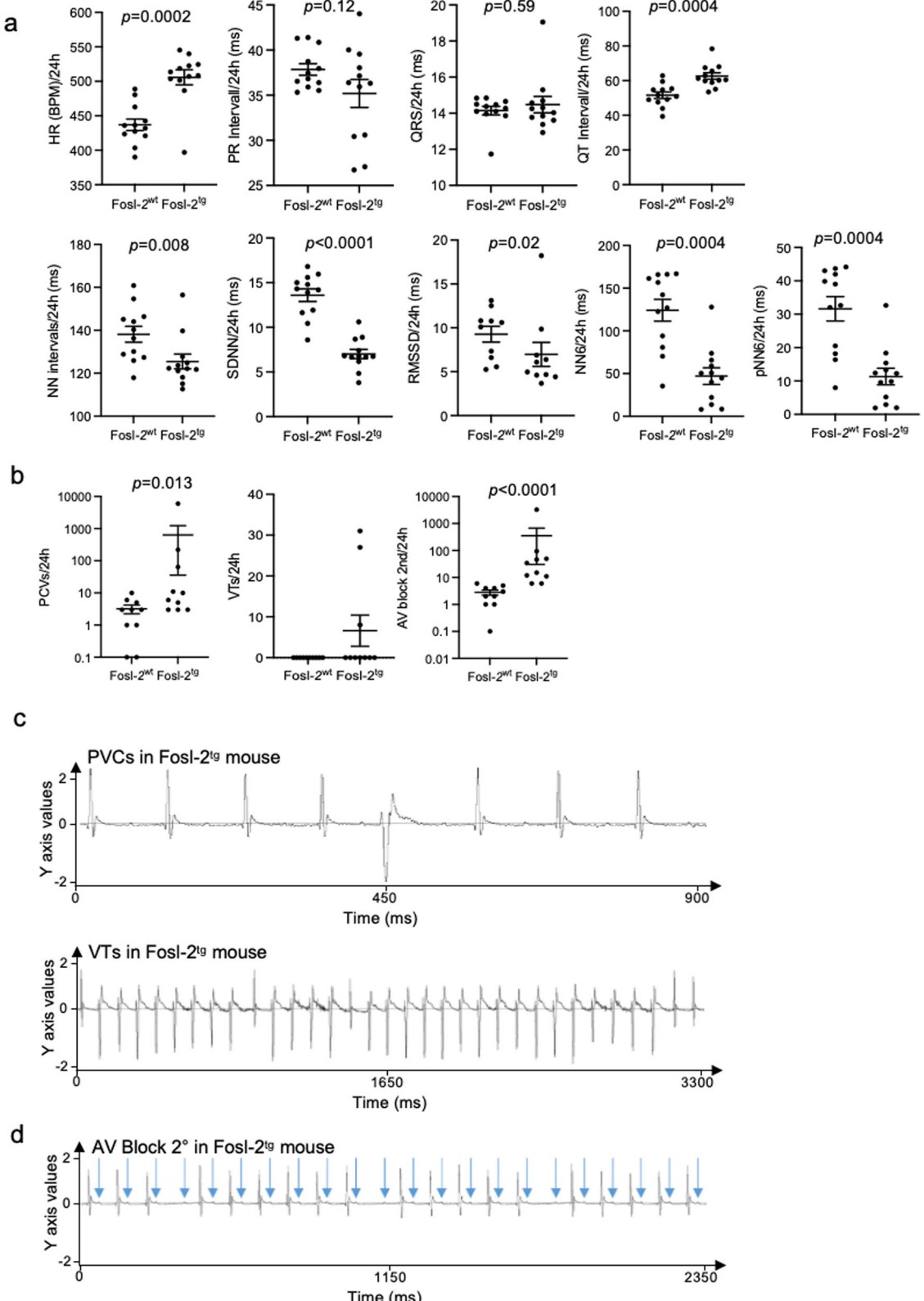

**Fig. 2 Fosl-2^tg mice show changes in ECG and in parasympathetic/sympathetic balance. a** Representative pictures of radiotelemetry ECG parameters and the heart rate variability parameters recorded within 24 h in Fosl-2^wt and Fosl-2^tg mice at week 16–22 (n = 12; unpaired t-test, mean ± SEM). **b** Analyses of PVC and VT events and AV blocks second-degree assessed by radiotelemetry and calculated as the average of 24 h of continuous ECG recording in awake Fosl-2^wt and Fosl-2^tg mice (n = 10, unpaired t-test, mean ± SEM). **c** Representative radiotelemetry ECG pictures of PVCs and VTs in Fosl-2^tg mouse. **d** AV block second-degree and representative pictures of AV blocks in Fosl-2^tg mouse. Arrows indicate P waves. HR heart rate, BPM beatings per minute, PVCs premature ventricular contractions, VTs ventricular tachycardias, AV block atrio-ventricular block, SDNN standard deviation of all NN intervals, RMSSD the square root of the root mean square of the sum of all differences between successive NN intervals, NN intervals interval between two normal heartbeats.

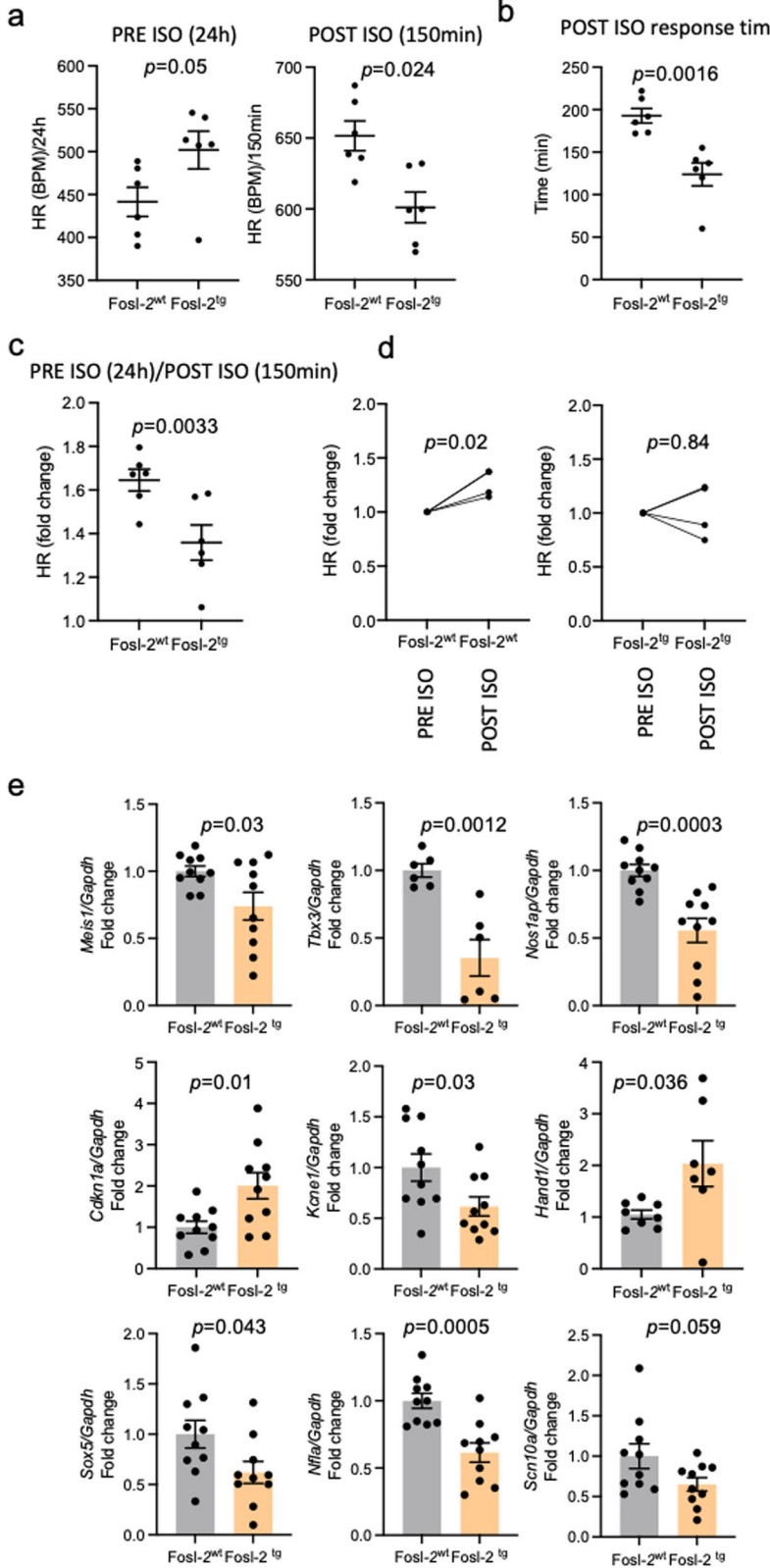

studies reported that up to one-third of SSc patients showed abnormal ECG parameters, which could lead to arrhythmias[18]. The most common ECG abnormalities and arrhythmic events in SSc patients include prolonged QT, AV blocks, PVCs and ventricular tachycardias[19–21]. We found that Fosl-2[tg] mice developed arrhythmias with AV-blocks, PVCs, VTs, and QT prolongation.

These data suggest that Fosl-2[tg] mice reflect key features of cardiac involvement in SSc patients. The reduction in heart rate variability (SDNN, RMSSD, NN, pNN6, and NN6) in combination with the increased baseline HR in Fosl-2[tg] mice indicated a higher sympathetic tone. Moreover, Fosl-2[tg] mice and isolated ex-vivo Fosl-2[tg] hearts showed impaired stress responses, further

**Fig. 3 Altered response to stress between Fosl-2^wt and Fosl-2^tg mice. a** Analyses of HR before isoproterenol (ISO) (PRE ISO, 24 h) and after ISO (POST ISO, 150 min) stimulation assessed by radiotelemetry and calculated as the average of continuous ECG recording in awake Fosl-2^wt and Fosl-2^tg mice ($n = 6$, unpaired $t$-test, mean ± SEM). **b** Analyses of time response assessed based of the evaluation of HR after ISO (POST ISO, 150 min) stimulation in awake Fosl-2^wt and Fosl-2^tg mice ($n = 6$, unpaired $t$-test, mean ± SEM). Mean of the HR measured before ISO (PRE ISO, 24 h) stimulation has been assessed as a time response threshold. **c** Analyses of HR fold changes before ISO (PRE ISO, 24 h) compared to after ISO (POST ISO, 150 min) stimulation in Fosl-2^wt or Fosl-2^tg mice ($n = 6$, unpaired $t$-test, mean ± SEM). **d** Analyses of HR fold changes before ISO (PRE ISO) and after ISO (POST ISO) stimulation of Fosl-2^wt and Fosl-2^tg hearts in ex-vivo Langendorff reperfusion system ($n = 4$, paired $t$-test). **e** Analyses of the gene regulatory elements in the cardiac conduction system analysed by qPCR in Fosl-2^wt and Fosl-2^tg hearts ($n = 6$–10, unpaired $t$-test, mean ± SEM).

indicating defects in the regulation by the autonomic nervous system. Importantly, SSc patients show an abnormal cardiac autonomic function evaluated by decreased heart rate variability, even in patients without cardiac symptoms[22–24].

Interstitial myocardial fibrosis has been recognized to trigger changes in the conduction system in SSc patients. Excessive number of cardiac fibroblasts and increased ECM deposition in cardiac tissue may lead to higher myocardial passive stiffness, separation of myocardial bundles, and impairment of electrical coupling between cells[25]. Interstitial fibrosis with reticular pattern is considered to have the greatest arrhythmogenic impact due to anisotropic re-entry based on loss of side-to-side electrical connections between cardiomyocytes[26]. We demonstrated increased numbers of cardiac fibroblasts in Fosl-2^tg mice and in SSc patient. Further, our data showed that these cells co-expressed myofibroblast markers (αSMA, ADAM12, periostin), exerted contraction properties and secreted collagens, which altogether clearly indicated a myofibroblast phenotype. Importantly, our data pointed to abnormalities in cardiac Cx43 and Cx40 (Gja5) in regions enriched in myofibroblast-like cells in Fosl-2^tg mice. Indeed, molecular mechanisms of fibrosis-induced arrhythmias are considered to involve Cx43 as the major gap junction protein expressed in the heart in both cardiomyocytes and fibroblasts[27] and is found mainly between atrial and ventricular cardiomyocytes, as well as in parts of the conduction system[28]. In heart failure heterogeneously redistributed Cx43 expression triggers arrhythmias, ventricular conduction and sudden arrhythmic death[29,30] and contributed to electrical uncoupling[31]. Patients with atrial fibrillation showed mutations in Cx40[32]. Cx40 like Cx43 plays also an important role in cardiac electrophysiology. Mice lacking Cx40 are characterized by slowed atrial conduction velocity and increased vulnerability of atrial arrhythmias[33]. Functional coupling between ventricular fibroblasts in the SAN is mediated by Cx40 in fibrotic cardiomyocyte-free areas[34]. Moreover, fibrotic lesions can mechanically and electrically separate surviving bundles of cardiomyocytes that can cause arrhythmia by creating areas of conduction block or reduce conductivity between cardiomyocytes[35]. In agreement with this notion, we demonstrated that Purkinje cells in Fosl-2^tg mice were surrounded by accumulated collagen and cardiac fibroblast that separated them from other cell types. Collectively these data suggest that in Fosl-2^tg mice, progressive cardiac fibrosis destructs physiological gap junction network and thereby triggers arrhythmic events.

Systemic inflammation correlates with disease severity in heart failure patients and promotes progression of the disease[36]. Fosl-2^tg mice displayed systemic and multiple organ inflammation, but the inflammatory response in the heart was rather low. Some studies have focused on mechanisms such as direct contribution of leukocytes in the electrical regulation of conducting cells or promotion of cardiac arrhythmias by autoantibodies and inflammatory cytokines that directly affected the function of ion channels on the surface of cardiomyocytes[37]. Our study, instead, demonstrated that the autoimmune response was necessary to trigger cardiac fibrogenesis and conduction system alterations in Fosl-2^tg mice. These results are consistent with several other studies, where mice with defects in adaptive immunity developed cardiac hypertrophy but were protected from ventricular dilation and adverse cardiac remodelling[38]. In fact, Fosl-2 overexpression in the absence of systemic inflammation (i.e. in Rag2^−/−Fosl-2^tg mice) neither promoted cardiac fibrosis, affected pathological distribution of cardiac Cx43, nor HR or QT intervals. Likewise, the RNAseq analysis confirmed that inflammation strongly affects cardiac fibroblasts by not only enhancing fibrotic changes, but also activating pathways implicated in arrhythmogenic alterations. Thus, these data point to importance of systemic inflammation in cardiac disease phenotype.

Fosl-2 overexpression in non-inflammatory conditions was not able to induce myocardial fibrosis and affect cardiac electrophysiology. However, RNAseq analysis revealed that Fosl-2 overexpression promoted to some extends profibrotic signature in cardiac fibroblasts. These data are in line with our recent findings showing that Fosl-2 knockdown in cardiac fibroblasts suppressed fibroblast-to-myofibroblast transition[39]. Importantly, we found that Fosl-2^tg fibroblasts expressed more Rgs4 in the myocardium, similarly to the failing human myocardium[40]. Rgs4 controls sinus rhythm by inhibiting parasympathetic signaling and acetylcholine-sensitive potassium current kinetics[41]. Rgs4 has been also found as a protein binding partner of Fosl-2 protein[42]. Therefore, the altered expression of Rgs4 in the myocardium of Fosl-2^tg mice may affect heart rhythm and parasympathetic signaling.

Taken together, we postulate a novel mechanism, by which Fosl-2 exaggerates arrhythmogenic cardiac fibrosis under immunofibrotic conditions. Therefore, pharmaceutical targeting of cardiac fibroblast activation and in particularly Fosl-2 might represent a promising alternative for the general anti-inflammatory treatment for SSc patients. Nevertheless, future studies are required to further understand the diversity and distinct activation properties of cardiac fibroblasts at different disease stages.

## Methods

**Murine and human specimens.** Fosl-2^tg mice were obtained from Sanofi Pharmaceuticals, Cambridge, USA (Supplementary Fig. 1) and backcrossed with C57Bl/6 mice for at least 10 generations. Genotype was confirmed by EGFP expression in blood samples. C57Bl6 mice were purchased from Charles River (stock number C57BL/6NCrl) and Rag2^−/− mice were purchased from Charles River and Jackson Laboratory respectively (stock number 008448). To generate Rag-2^−/−Fosl-2^tg, Fosl-2^tg mice were crossed with Rag-2^−/− mice. Genotype was confirmed by EGFP expression in blood samples. All mice were housed in pathogen free conditions. For all experiments in the manuscript, age-matched male (60%) and female mice (40%) were used, unless differently stated. Number of mice are indicated in the single experiments. Animal experiments were performed in accordance with the Swiss federal law and with the Guide for the Care and Use of Laboratory Animals published by the US National Institutes of Health (NIH Publication, 8th Edition, 2011). Cantonal Veterinary Office Zurich had approved all animal experiments (ZH28/2015, ZH007/2019).

Human endomyocardial biopsies (hEMBs) from SSc patients and control hEMBs from the patients with healed myocarditis (Supplementary Tables 1 and 2) were provided by the Cardiopathology, Institute for Pathology and Neuropathology, University Hospital Tubingen, Germany. The experiments with re-usage of human material were approved by Swissethics (BASEC-Nr. 2019-00058) and were performed in conformity with the principles outlined in the Declaration of Helsinki.

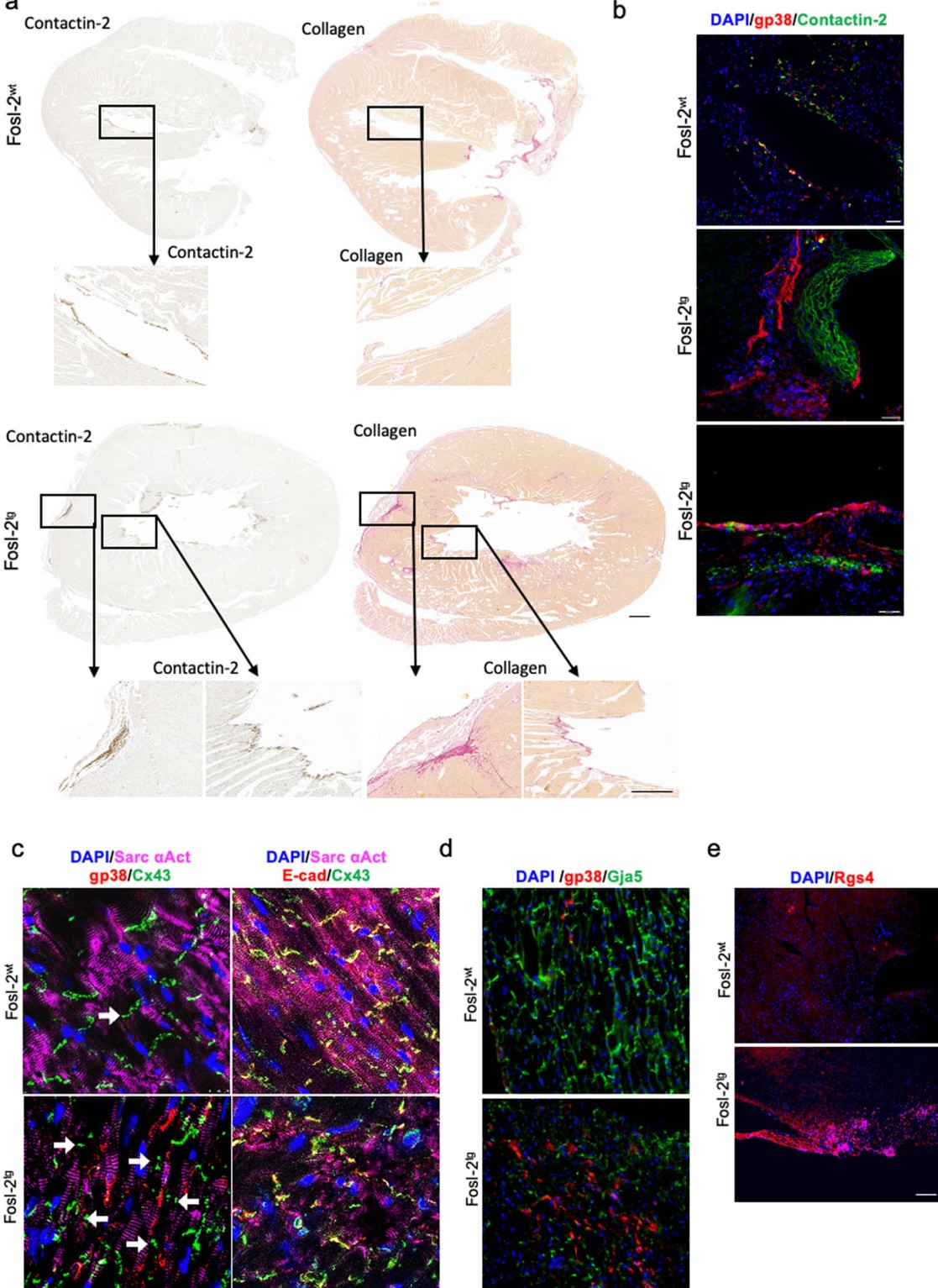

**Fig. 4 Alteration in the gap-junction and association of cardiac conduction system with fibrosis in Fosl-2^tg mice. a** Representative pictures of contactin-2 IHC and Sirius Red staining in the myocardial sections from 16 to 22-week-old Fosl-2^wt and Fosl-2^tg mice ($n = 6$). Scale bar: 500 μm, insert scale bar: 50 μm. **b** Representative pictures of contactin-2 (green) and gp38 (red) immunofluorescence counterstained for nuclei with DAPI (blue) in the myocardial sections from 16 to 22-week-old Fosl-2^wt and Fosl-2^tg mice ($n = 6$). Scale bars: 100 μm. **c** Representative pictures of connexin-43 (Cx43, green) or E-cadherin (red), gp38 (red) and sacromeric α-actinin (Sarc αAct, pink) immunofluorescence counterstained for nuclei with DAPI (blue) in the myocardial sections from 16–22-week-old Fosl-2^wt and Fosl-2^tg mice ($n = 6$). Scale bars: 50 μm. **d** Representative pictures of connexin-40 (Gja5, green) and gp38 (red) immunofluorescence counterstained for nuclei with DAPI (blue) in the myocardial sections from 16 to 22-week-old Fosl-2^wt and Fosl-2^tg mice ($n = 6$). Scale bars: 50 μm. **e** Representative pictures of Rgs4 (red) immunofluorescence counterstained for nuclei with DAPI (blue) in the myocardial sections from 16 to 22-week-old Fosl-2^wt and Fosl-2^tg mice ($n = 4$). Scale bars: 50 μm.

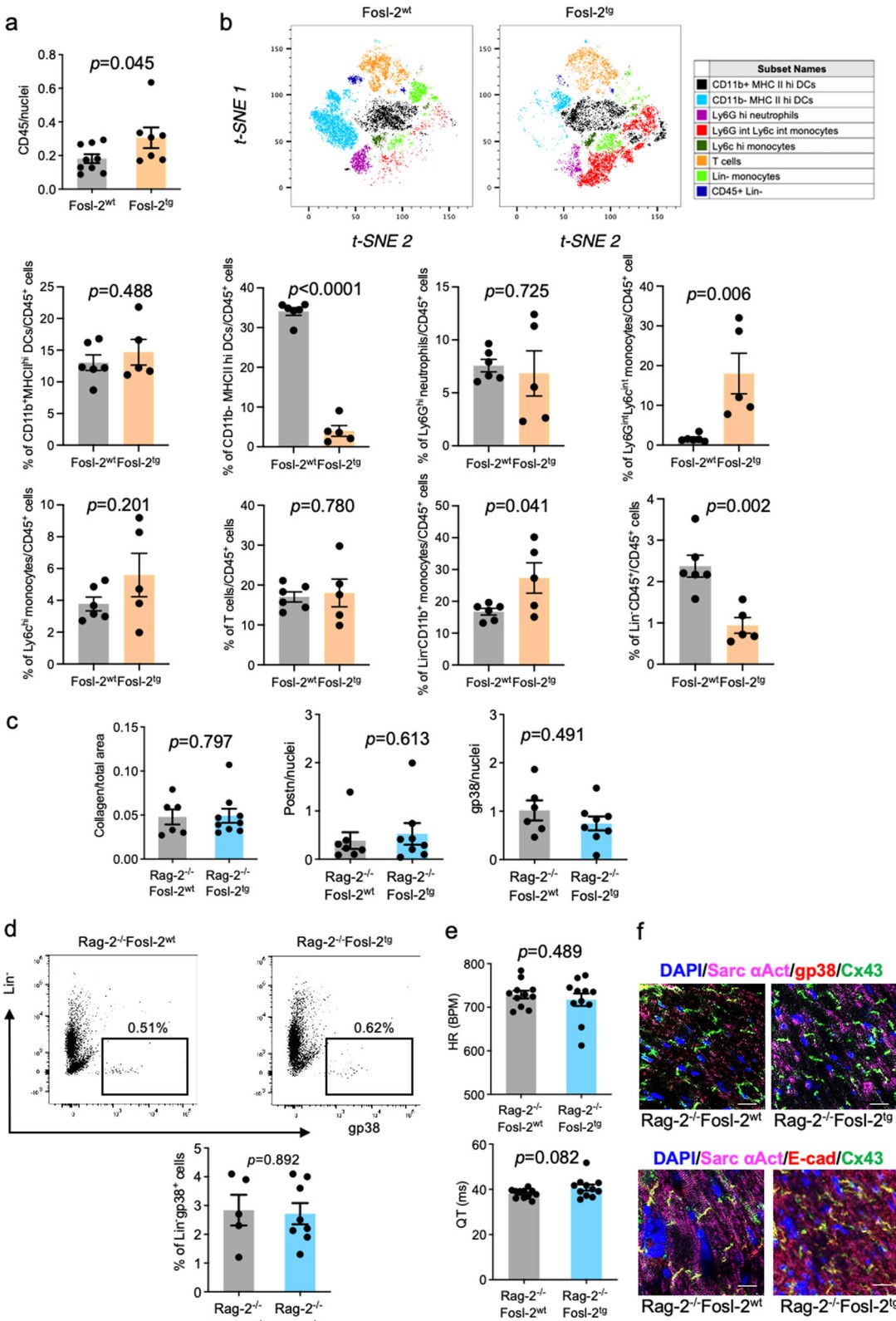

**Radiotelemetry**. DSI PhysioTel® Implantable Telemetry was performed. 18-20-week-old mice were weighed and anesthetized by inhalation with 5% isoflurane for induction and 2.5% for maintenance of general anaesthesia. Radiotransmitters were implanted by surgery. The analgesic (Metamizol, 200 mg/kg) was administrated by subcutaneous injection 30 minutes before the surgery for radiotransmitter implantation. The positive lead was positioned in the left caudal rib region and the negative lead under the right pectoral muscle. All animals received a local anaesthetic applied to the abdominal wall before skin closure

(Bupivacaine 0.25%, intra-incisional, total dose not exceeding 8 mg/kg). Additional analgesia was administrated for at least 3 days after surgery with subcutaneous injection of Metamizol (200 mg/kg) and warmed Ringer-Lactate solution twice daily. Animals were housed at 30 °C for at least 3–4 post-operative days and let to recover for 1 week. Next, ECG was recorded continuously up to 3 weeks, respectively at week 18–19, 19–20, and 20–21 of age. ECGs were recorded in freely moving unrestrained mice. To analyze the in vivo stress-response we performed double injections (with 30 min interval) of isoproterenol

**Fig. 5 Inflammation in the heart of Fosl-2$^{tg}$ mice and myocardial phenotype in Rag2$^{-/-}$Fosl-2$^{tg}$ mice. a** Quantification of CD45 IHC staining on myocardial sections from 16 ti 22-week-old mice ($n = 7$–9, Mann–Whitney test, mean ± SEM). **b** Representative flow cytometry t-SNE2 blots and corresponding quantification of immune cell populations (Ly6G high (hi) neutrophils, Ly6G intermediate (int) monocytes, Ly6c int monocytes, Lin$^-$CD11b$^+$ monocytes, CD11b$^+$MHCII hi dendritic cells (DCs), CD11b$^-$MHCII hi DCs, Lin$^-$CD45$^+$ cells and T cells) from the myocardium of 16–22-week-old mice ($n = 5$–6, unpaired $t$-test). Gating strategy is presented in Supplementary Fig. 10. **c** Quantification of Sirius Red, periostin (Postn) and gp38 IHC staining of myocardial sections from 16 to 22-week-old mice ($n = 6$–8, Mann–Whitney test, mean ± SEM). **d** Representative flow cytometry pictures and corresponding quantification of Lin$^-$gp38$^+$ fibroblast frequency from the myocardium of 16–22-week-old mice ($n = 5$–8, Mann–Whitney test, mean ± SEM). **e** HR and QT intervals recorded in 16–22-week-old mice ($n = 11$, unpaired $t$-test, mean ± SEM). **f** Representative pictures of connexin-43 (Cx43, green), gp38 (red) or E-cadherin (red) and sacromeric α-actinin (Sarc αAct, pink) immunofluorescence counterstained for nuclei with DAPI (blue) in the myocardial sections from 16 to 22-week-old Fosl-2$^{wt}$ and Fosl-2$^{tg}$ mice ($n = 5$). Scale bars: 50 μm.

---

(intraperitoneally 2 mg/kg). The analyses were performed with the Ponemah Physiology Platform System.

**In vitro measurements**. Analyzed mice were of mixed gender. Mice were sacrificed and perfused with cold PBS via the left ventricle. Hearts were harvested, deprived of the atria and weighed. Tibias were harvested and cleaned from muscles and fat in order to measure the length.

**Preparation of cardiac stromal single cell suspension**. We previously described a new, simple method to isolate cardiac fibroblasts from the heart C57Bl/6 mice[15]. Briefly, hearts were harvested after cold PBS perfusion through the left ventricle and kept on ice before further proceeding. Tissues were mechanically and enzymatically dissociated using magnetic beads (VWR) in 0.025 mg/mL of Liberase TM (Collagenase I and II, Roche) solution in pure Dulbecco's modified Eagle's medium (DMEM, Gibco) complemented with DNAse (40 µg/mL, Stemcell Technologies) on a magnetic shaker for 45 min at 37 °C. Cells were isolated by filtering the resulting cell suspension through a 70 µm cell strainer followed by a 500-rpm centrifugation to allow the deprivation of cardiomyocytes. Successively, supernatants were passed through a 40 µm cell strainer, centrifuged at 1400 rpm for 4′ and washed with PBS. Gp38$^+$ cells were pre-selected with anti-gp38-APC staining (eBioscience) followed by magnetic anti-APC-microbeads (Miltenyi Biotec) staining and autoMACS® Pro (Miltenyi Biotec) separation. Finally, enriched gp38$^+$ cells were further FACS-sorted or used for cell culture and in vitro assays.

Foetal human cardiac fibroblasts were purchased from Sigma (Cell Applications), and cells from passages 10–15 were used.

**Deep RNA sequencing**. RNA-sequencing was performed on total RNA extracted from FACS-sorted Lin$^-$gp38$^+$ fibroblasts isolated from the hearts of Rag-2$^{-/-}$Fosl-2$^{wt}$, Rag-2$^{-/-}$Fosl-2$^{tg}$, and Fosl-2$^{tg}$ mice (3 mice in each mouse strain, each in 2 technical replicates). The first analysis performed to identify the significantly deregulated genes between Lin$^-$gp38$^+$Rag-2$^{-/-}$Fosl-2$^{tg}$ and Lin$^-$gp38$^+$Fosl-2$^{tg}$, served to investigate the role of inflammation. Significantly deregulated genes were identified computing the Log2 fold change of the normalized mean hit counts with the formula Log2(Lin$^-$gp38$^+$Fosl-2$^{tg}$ mean normalized counts/Lin$^-$gp38$^+$Rag-2$^{-/-}$Fosl-2$^{tg}$ mean normalized counts) = Log2 fold change. $P$-values were calculated with the Wald test and adjusted with the Benjamini–Hochberg analysis. Differential gene expression was considered significant with Benjamini-Hochberg adjusted $p$-value ≤ 0.05. Gene expression in Lin$^-$gp38$^+$Fosl-2$^{tg}$ was considered downregulated with Log2 fold change ≤ −1 and upregulated with Log2 fold change ≥1 as compared to the control group Lin$^-$gp38$^+$Rag-2$^{-/-}$Fosl-2$^{tg}$. Next, the significant upregulated genes uniquely deregulated in the comparison between Lin$^-$gp38$^+$Rag-2$^{-/-}$Fosl-2$^{tg}$ and Lin$^-$gp38$^+$Fosl-2$^{tg}$, were used as input to the comprehensive gene set enrichment web server EnrichR. In parallel, significantly deregulated genes between Lin$^-$gp38$^+$Rag-2$^{-/-}$Fosl-2$^{wt}$ and Lin$^-$gp38$^+$Rag-2$^{-/-}$Fosl-2$^{tg}$ served to investigate the role of Fosl-2. In this analysis, the formula Log2(Lin$^-$gp38$^+$Rag-2$^{-/-}$Fosl-2$^{tg}$ mean normalized counts/Lin$^-$gp38$^+$Rag-2$^{-/-}$Fosl-2$^{wt}$ mean normalized counts) = Log2 fold change was used. Differential gene expression was considered significant with Benjamini–Hochberg adjusted $p$-value ≤ 0.05. Gene expression in Lin$^-$gp38$^+$Rag-2$^{-/-}$Fosl-2$^{tg}$ was considered downregulated with Log2 fold change ≤ −1 and upregulated with Log2 fold change ≥ 1 as compared to the control group Lin$^-$gp38$^+$Rag-2$^{-/-}$Fosl-2$^{wt}$. Subsequently, the significant upregulated genes uniquely deregulated in the comparison between Lin$^-$gp38$^+$Rag-2$^{-/-}$Fosl-2$^{wt}$ and Lin$^-$gp38$^+$Rag-2$^{-/-}$Fosl-2$^{tg}$ were used as input in EnrichR. RNA sequencing data were submitted to the ArrayExpress repository under the accession code E-MTAB-12271.

**RNA extraction**. Cells were harvested by trypsinization, washed with PBS and mixed in RNA lysis buffer (Zymo Research). Cardiac tissue was lysed in 600 µL of RNA lysis buffer (Zymo) and homogenized using TissueLyserII (Qiagen). Homogenate was spun down for 2 min at 15000 g and the supernatant was processed according to the manufacturer's instructions. Zymo Quick-RNA MicroPrep isolation kit was used to extract total RNA. 100% ethanol was added to the lysates and samples were than processed on the columns. DNA contamination was removed by DNase I digestion. RNA was washed and eluted in 20 µl of water. RNA concentration and purity were assessed with Nanodrop 1000 (Thermo Fisher Scientific).

**Reverse transcription-quantitative polymerase chain reaction**. Reverse transcription of 125 ng of RNA was performed using the Transcriptor First Strand cDNA Synthesis kit (Roche) and random hexamers, according to the manufacturer's protocol. Briefly, RNA mixed with random hexamers was denatured at 65 °C for 10 min. Afterwards, 5x reaction buffer, 10 mM dNTPs, reverse transcriptase and RNase inhibitor were added to the reaction. The cDNA synthesis program was the following: 10 min at 25 °C, 60 min at 50 °C, followed by 5 min at 85 °C. Subsequent qPCRs were performed with 2x SYBR Green master mix (Promega) on an Agilent Technologies Stratagene Mx3005P qPCR system. The amplification program consisted in a first step at 95 °C for 10 min, then 40 cycles of 95 °C for 30 s, and 60 °C for 60 s. A last amplification cycle was added to record the melting curve. Each sample was amplified in duplicate together with no retro-transcriptase controls and no template controls. Glyceraldehyde 3-phosphate dehydrogenase (GAPDH) was used as reference housekeeping gene. Relative expression levels were calculated with the 2$^{-\Delta\Delta Ct}$ method. Primers used for reverse transcription-quantitative polymerase chain reaction analyses in murine and human cells are listed in Suppl. Table 4 and 5, respectively.

**ELISA**. Pro-collagen I levels in cell culture supernatants were measured according to the manufacturer's protocol using mouse pro-collagen I alpha 1 ELISA (Abcam) and human pro-collagen I alpha 1 ELISA (R&D Systems). Supernatants were diluted 1:50 in starvation medium (1% serum DMEM) and 50 µl of diluted sample was added to one well of a 96-well plate pre-coated with immobilizing anti-pro-collagen-1 antibodies. Antibody Cocktail solution containing capture and detector antibodies was added to each sample and incubated for 1 h at RT. Finally, TMB (3,3′,5,5′-Tetramethylbenzidin) substrate was added and incubated for 10 min at RT in the dark. The reaction was stopped with stop solution. Absorbance was measured at 450 nm in the Synergy HT (Biotek) plate reader. All samples were measured in duplicates. Mean absorbance was calculated and concentrations/total amount were determined using the respective standard curve (from 0 to 2.5 µg/mL).

**Flow cytometry and cell sorting**. Sigle cell suspension was prepared as described in the previous section. Next, a blocking step was performed with anti-mouse CD16/CD32 (Fc blocking) (Clone 93, Thermo Fisher Scientific) for 10 minutes, followed by incubation on ice for 30 minutes with the appropriate combination of fluorochrome-conjugated antibodies. Antibodies used for flow cytometry experiments are listed in Suppl. Table 6 (dilutions 1:300–1:600). Cells were analysed with the BD LSR Fortessa FACS (BD Biosciences) and sorted with FACS Aria III 4 L (BD Biosciences). Flow cytometry data including t-SNE analysis were performed using the FlowJo v.10 software.

**Protein extraction and western blotting**. Proteins were extracted with RIPA Buffer (Sigma-Aldrich) complemented with protease inhibitor cocktail (Complete ULTRA Tablets, Roche) and phosphatase inhibitors (PhosphoStop, Roche) from cultured cells or homogenised heart tissues: cultured cells were trypsinised for 5 min at 37 °C, washed and re-suspended in RIPA Buffer. Heart were homogenised in RIPA Buffer with Minilys by Bertin Technologies and 30 s homogenizing cycle interspersed by 30 second break for a total of 3 or 4 cycles until tissue was completely disrupted. Protein concentration was quantified by colorimetric BCA method according to the manufacturer's protocol (Thermo Fisher Scientific). SDS-PAGE electrophoresis and wet-transfer method were used to separate and transfer proteins on nitrocellulose membranes, then incubated 45 minutes in blocking solution Tris buffered saline and Tween-20 (Thermo Fisher Scientific) (TBST) containing 5% milk. Membranes were probed overnight with the primary antibodies Suppl. Table 5). GAPDH was used as loading control. HRP-conjugated secondary antibodies were used for detection with ECL substrate (SuperSignal West Pico Plus, Thermo Fisher Scientific) and development on the Fusion Fx (Vilber). PageRuler Plus protein ladder (Thermo Fisher Scientific) was used for protein size evaluation. Densitometric analyses were conducted with ImageJ 1.47t. Fold changes were computed after normalization to GAPDH densitometry.

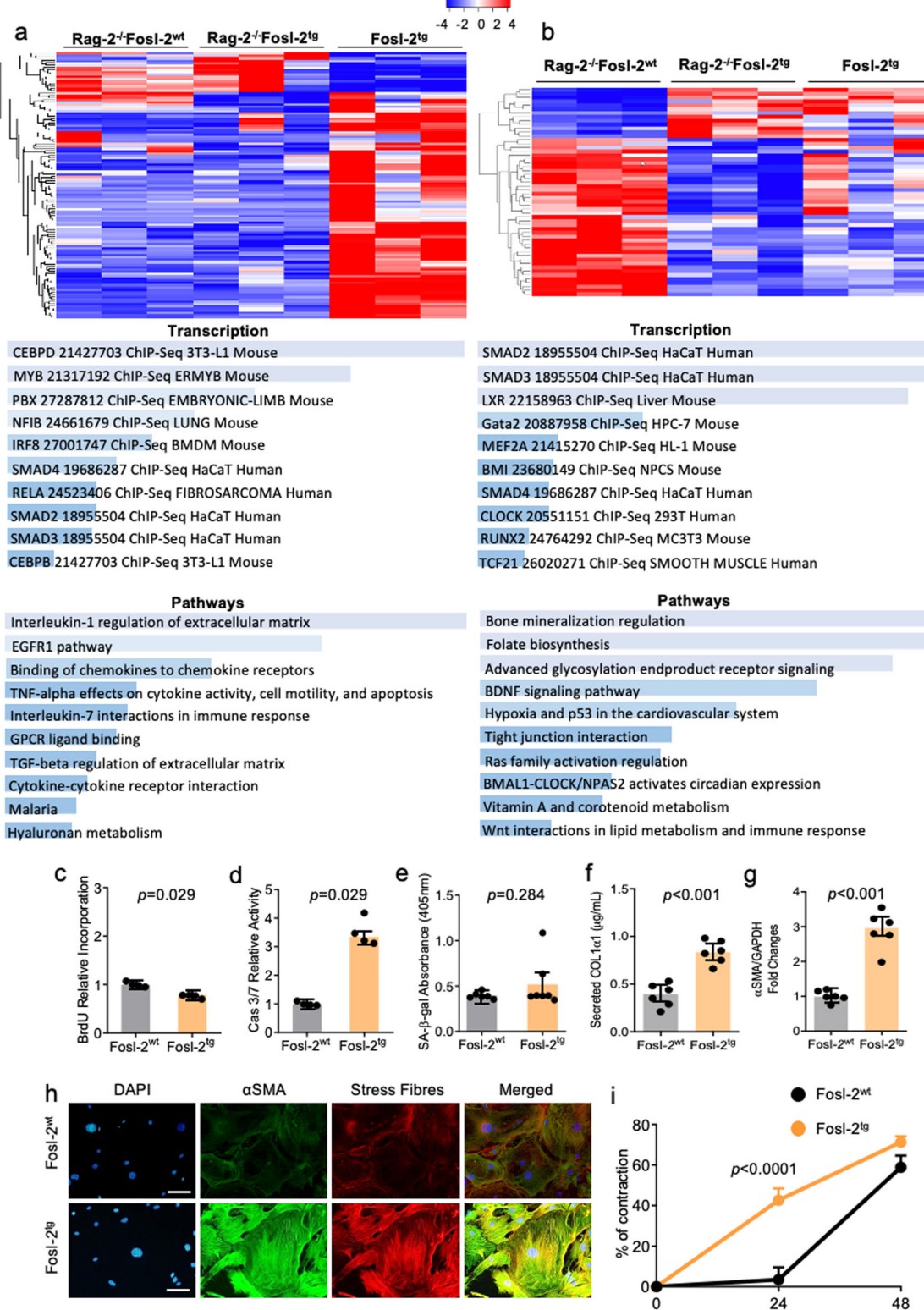

**Cell culture**. Lin⁻gp38+ cells were isolated from Fosl-2ʷᵗ, Fosl-2ᵗᵍ, Rag-2⁻/⁻Fosl-2ʷᵗ, and Rag-2⁻/⁻Fosl-2ᵗᵍ mice. Primary human foetal cardiac fibroblasts were purchased from Cell Applications inc. Both murine and human cells were cultured in DMEM (Gibco) containing 10% foetal bovine serum (FBS), 50 U/mL penicillin, 50 μg/mL streptomycin (Gibco), and 50 mM β-mercaptoethanol (Thermo Fisher Scientific). In selected experiments, cells were stimulated with 10 ng/mL recombinant TGF-β (Peprotech) and 10 ng/mL LPS (InvivoGen - Labforce AG). Murine Lin⁻gp38+ cells from passages 2 to 5 and human cardiac fibroblasts from passages 5 to 15 in monolayer culture were used for the experiments.

**Transient Fosl-2 overexpression**. Human cardiac fibroblasts were cultured in normal medium and used at passage 6-14. Overexpression of human Fosl-2 was performed using pcDNA3.1(+) vector carrying the gene sequence of human Fosl-2 (Accession Number: XM_006711976.2), purchased from GenScript. Cells were transfected 24 h after seeding, using 3.75 μl/mL of Lipofectamine 2000 (Thermo Fisher Scientific) and harvested at 24 and 48 h after transfection. pcDNA3.1(+) vector deprived of Fosl-2 gene sequence was used as reference control.

**Fig. 6 Lin⁻gp38⁺Fosl-2ᵗᵍ fibroblasts are committed to myofibroblast phenotype. a** Heatmap of significantly deregulated genes between Lin⁻gp38⁺Rag2⁻/⁻Fosl-2ᵗᵍ and Lin⁻gp38⁺Fosl-2ᵗᵍ expressed in Transcripts Per Million (TPM) for Rag2⁻/⁻Fosl-2ʷᵗ, Rag2⁻/⁻Fosl-2ᵗᵍ and Fosl-2ᵗᵍ groups. Analysis was performed with the comprehensive gene set enrichment web server EnrichR for Transcription (database ChEA 2016) and Pathways (database BioPlanet 2016) areas of significantly deregulated genes uniquely upregulated in Lin⁻gp38⁺Fosl-2tg vs. Lin⁻gp38⁺Rag2⁻/⁻Fosl-2ᵗᵍ fibroblasts. Cardiac fibroblasts were isolated for the hearts of 16-20-week-old mice. **b** Heatmap of significantly deregulated genes between Lin⁻gp38⁺Rag2⁻/⁻Fosl-2ʷᵗ and Lin⁻gp38⁺Rag2⁻/⁻Fosl-2ᵗᵍ expressed in TPM for Rag2⁻/⁻Fosl-2ʷᵗ, Rag2⁻/⁻Fosl-2ᵗᵍ, and Fosl-2ᵗᵍ groups. Analysis as in **a** of significantly deregulated genes uniquely upregulated in Lin⁻gp38⁺Rag2⁻/⁻Fosl-2ᵗᵍ vs. Lin⁻gp38⁺Rag2⁻/⁻Fosl-2ʷᵗ. Quantification of: **c** 5-bromo-2′-deoxyuridine (BrdU) proliferation assay, **d** Caspase 3/7 activity assay, **e** Quantification of Senescence Associated β-galactosidase (SA-β-gal) activity of cardiac Lin⁻gp38⁺ fibroblasts from 16 to 22-week-old Fosl-2ʷᵗ and Fosl-2ᵗᵍ mice (n = 4–6, unpaired t-test, for all mean ± SEM). **f** ELISA assay for quantification of pro-collagen I levels (n = 6, unpaired t-test, mean ± SEM), **g** densitometric analysis of αSMA protein content (n = 6, unpaired t-test, mean ± SEM), **h** double IF staining for αSMA and stress fibres, nuclei are stained with DAPI (blue) (n = 4, scale bar = 50 μm), and contraction assay (**i**, n = 3, two-way ANOVA test with Sidak's multiple comparisons test, mean ± SEM) in Lin⁻gp38⁺Fosl-2ʷᵗ and Lin⁻gp38⁺Fosl-2ᵗᵍ fibroblasts.

**Immunofluorescent staining.** OCT-embedded heart cryosections previously incubated in 30% Sucrose-PBS overnight were fixed with methanol:acetone (7:3) (both Sigma) for 10 minutes at −20 °C. First, cardiac sections were permeabilised with 0.1% TritonX-100 (Sigma-Aldrich) in PBS complemented with 1% serum for 10 minutes at RT, then washed three times in PBS after which unspecific binding sites were blocked using 10% serum in PBS for 20 minutes. Next, the primary antibodies (Suppl. Table 5) were incubated overnight at 4 °C. The next day, slides were washed three times in PBS and secondary antibody (1:400) were incubated for 45 min at RT. Slides were further washed in PBS and then incubated with anti-hamster IgG gp38 (1:100) for 2 h at RT. AlexaFluor 546 anti-hamster IgG (1:400) was dispensed on cardiac sections and left for 45 min at RT. Finally, after 3 washings in PBS 4′,6-diamidino-2-phenylindole (DAPI 1 μg/ml, Roche) was used to label nuclei. Stained heart sections were imaged with the inverted confocal microscope Leica DMI6000 AFC, Model SP8.

For αSMA immunofluorescence staining combined with Phalloidin–Tetramethylrhodamine B isothiocyanate (Sigma) staining, cells were seeded in 8-chamber glass slides (Lab-Tec) at the density of 10,000 cells/well. After 24 h of culture, the cells were stimulated with or without TGF-β for 72 hours. Culture medium was removed ad cells were washed twice with PBS. Next, a fixation step in ice-cold methanol-acetone (7:3, both Sigma-Aldrich) for 10 min at −20 °C was performed. Cells were them washed three times with PBS and unspecific binding sites were blocked with 10% FBS in PBS for 20 minutes at RT. Subsequently, primary anti-αSMA antibody (1:100) incubation was performed for 1 h at RT followed by three washes in PBS. Then, cells were stained with the secondary antibody (chicken anti-mouse, 1:400, Thermo Fischer Scientific) for 45 minutes at RT. For stress fibres staining, cells were fixed in 4% paraformaldehyde (PFA, Sigma-Aldrich) for 5 minutes, washed three times in PBS, permeabilized in PBS 0.1% TritonX-100 (Sigma-Aldrich) and washed again three times in PBS. Finally, cells were stained with 50 μg/ml of fluorescent labelled phalloidin (Sigma-Aldrich) for 40 minutes at RT. After washing, nuclei were counterstained with DAPI solution (1 μg/ml, Roche). Images were acquired with an Olympus BX53 microscope equipped with a DP80 camera.

**Immunohistochemistry.** Hearts isolated from sacrificed mice after cold PBS perfusion were fixed with 4% formalin (Sigma) for 12 hours and paraffin embedded. Direct Red Sirius Red (Sigma-Aldrich) staining was used to detect collagen fibres. For immunohistochemistry, antigen retrieval was performed on rehydrated sections. In brief, sections were boiled in citrate buffer (10 mM Citrate, 0.05% tween, pH = 6, Sigma) with a microwave, and further incubated at 95 °C in an oven for 15 min. Sections were left to cool down and washed in PBS. To block endogenous peroxidases, sections were incubated 15 minutes in 3% H₂O₂ solution and rinsed twice in PBS-Tween for 5 min. To block unspecific antibody binding, sections were incubated for one hour in blocking solution (10% Goat serum in Background Reducing Antibody Diluent from Dako). Endogenous biotin was blocked using Avidin-Biotin Block kit (from Vector Laboratories). The primary antibody was incubated overnight at 4 °C followed by two washings in PBS-Tween for 5 min. The proper biotinylated secondary antibody (Vector Laboratories) was incubated for 30 min at room temperature. After washing twice with PBS-Tween for 5 min, sections were incubated for 30 min with VECTASTAIN Elite ABC solution (Vector Laboratories) and washed twice in PBS for 5 min. Staining was developed using DAB (from Vector Laboratories) followed by a counterstaining of nuclei for 1 minute in Mayer's haematoxylin solution (J.T Baker). Used primary antibodies are listed in Suppl. Table 6. Biotin-labelled goat anti-rat, goat-anti-rabbit, or goat anti-hamster secondary antibody were all purchased from Vector Laboratories.

**Immunohistochemistry quantification.** Hearts processed for immunohistochemistry were cut transversely into 2 or 3 sections. The entire slide containing all cardiac sections from one mouse was scanned using Slidescanner Zeiss Axio Scan microscope. ImageJ software was used to quantify signals from the sections expressed in areas. For Sirius Red quantification, colour deconvolution plugin "FastRed FastBlue" was performed. The same threshold for colour intensity was applied to all images to obtain the number of positive pixels. Separated images were used to calculate the area of collagen and the total area of the section. The value for each mouse was calculated as the average of collagen/total area ratio of three single sections. For all the other DAB IHC staining, "H DAB" colour deconvolution plugin was used. Separated images were used to calculate the area of the DAB signal and the area of the nuclear staining. The value for each mouse was calculated as the average of DAB signal/nuclei ratio of two or three single sections.

**Cell contraction assay.** Contraction Assay Kit (Cell Biolabs) was used to assess contractile properties of cultured cells following the manufacturer's protocol. Briefly, Lin⁻gp38⁺ cells were cultivated with or without 10 ng/mL TGF-β for 72 h. Primary human cardiac fibroblasts were cultivated with or without 10 ng/mL TGF-β in addition to the inhibitors SD208 or A83-01 (both Tocris) for 72 h. Next, cells were detached by trypsinization and re-suspend in proper medium at 2–5 × 10⁶ cells/mL. Two parts of cell suspension and eight parts of cold Collagen Gel Working Solution composed of collagen, DMEM and neutralization agent were mixed in order to plate and 0.25 mL of the cell-collagen mixture per well in a 48-well plate. Collagen was allowed to polymerize for 1 h at 37 °C. After that 1 mL of culture medium was added atop each collagen gel lattice. Each condition was analysed in triplicates or quadruplicates. Images were taken at time 0, 24, 48, and 72 h after seeding. Areas of the gels were measured by ImageJ. Percentage of contraction of all conditions was measured compared to the average of unstimulated cells at time 0 set as 100%.

**Bromodeoxyuridine ELISA proliferation assay.** Lin⁻gp38⁺ cells were seeded in 96-well plates (10,000 cells/well). 24 h after seeding, bromodeoxyuridine (BrdU) was added to the culture medium for the next 24 h. BrdU incorporation was measured by ELISA. Briefly, after removing labelling medium, cells were fixed with a fixative/denaturing solution for 30minutesand further incubated with anti-BrdU-POD antibodies for 90 min at RT. After several washes with PBS, substrate solution was added to the wells. Absorbance was measured at 450 nm with correction at 690 nm with the Synergy HT microplate reader (BioTek). Each sample was measured in quadruplicates. Mean absorbance was calculated and unstimulated cells served as control. All experimental conditions were calculated as fold change to controls.

**Caspase Glo 3/7 assay.** For apoptosis assay, Caspase 3/7 activity was detected by Caspase-Glo® 3/7 Assay Systems (Promega) following the manufacturer's protocol. Lin⁻gp38⁺ cells were seeded in 96-well plates (10,000 cells/well). 24 h after seeding, freshly prepared Caspase-Glo® reagent was prepared and distributed to the wells followed by an incubation in the dark for 90 min at RT. Luminescent signals were measured with the Synergy HT microplate reader (BioTek). All conditions were analysed in quadruplicates. Mean luminescence was calculated and unstimulated cells served as controls. All experimental conditions were calculated as fold change to controls.

**Senescence Associated-β-Galactosidase assay for cell senescence.** The Senescence Associated-β-Galactosidase Staining Kit (Cell Signalling) was used to detect β-galactosidase activity at pH 6, a known characteristic of senescent cells. Lin⁻gp38⁺ cells isolated from the myocardium of 7- and 16–22-week-old mice were seeded and cultured for 4 days in normal medium before the analysis. Protocol was performed according to the manufacturer's protocol. Shortly, growth media was removed from cultured cells and a washing step with PBS was performed. Cells were fixed with Fixative Solution 10–15′ at RT and then washed twice with PBS. β-Galactosidase Staining Solution was added to each well and left to incubate at 37 °C overnight in a dry incubator. β-galactosidase staining was then checked under an Olympus inverted IX81 microscope equipped with F-view camera.

**Electrocardiogram in conscious mice.** Heart rate (HR) of Rag-2⁻/⁻ Fosl-2ʷᵗ and Rag-2⁻/⁻ Fosl-2ᵗᵍ mice was recorded in conscious mice at the indicated ages with non-invasive ECGenie apparatus platform (eMouseSpecifics, Inc., Boston, MA, USA). The device acquires signals through disposable footpad electrodes located in the floor of a recording platform. Mice were trained every day 10 min for 3 days

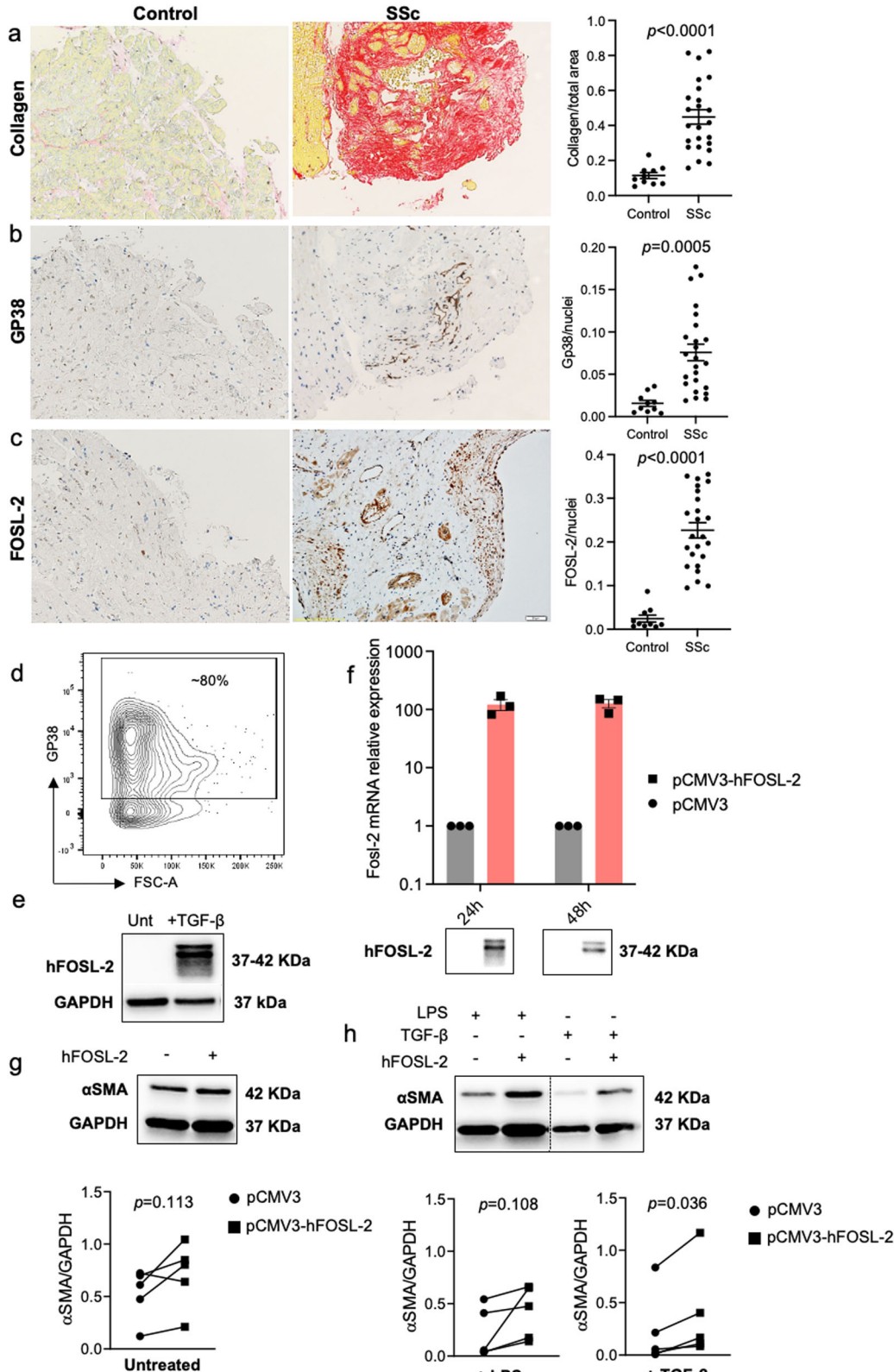

**Fig. 7 GP38 and FOSL-2 expression in human specimens.** Representative pictures and relative quantification of Sirius red (**a**), GP38 IHC (**b**), and FOSL-2 IHC (**c**) staining of EMBs from HF/SSc patients and donors with healed myocarditis ($n = 10$–24, unpaired $t$-test, mean ± SEM, scale bar: 20 μm). **d** Representative flow cytometry plot of GP38-expressing human foetal cardiac fibroblasts (fCFs). Panel **e** illustrates representative immunoblots of FOSL-2 in fCFs in response to treatment with TGF-β (10 ng/mL) for 24 h. **f** hFOSL-2 overexpression in fCFs assessed by qPCR and WB analysis ($n = 3$, mean ± SEM). Panels **g** and **h** show representative immunoblots and WB analysis of fCFs transfected with pCMV control and pCMV-FOSL-2 vectors at basal condition in 10% FBS DMEM (F) and after stimulation with TGF-β or LPS for 24 h (G) (all *paired t-test*, $n = 5$).

 

prior the analysis. On the 4th day, mice were allowed to acclimatize for ~10 min, and raw data were recorded for 10 s. Raw ECG signals were analysed using EzCG analysis software from eMouseSpecifics.

**Echocardiography**. Echocardiography was performed in 20-week-old males using a digital small animal ultrasound system (Vevo 2100 Imaging System, VisualSonics, Toronto, Canada) with a 18–38 Mhz linear-array probe. Anaesthesia was induced with 5% isoflurane and keep at 2–2.5% during echocardiography. Standardized two-dimensional long- and short axis views were obtained from all animals to assess LV size and function. LV ejection fraction (EF) was calculated as ([end-diastolic volume—end-systolic volume]/end-diastolic volume) × 100. Analysis of all images was performed offline by the semi-automated strain measurements using the Vevo2100 Software Version 1.6. All the reported parameters were calculated in M-Mode.

**Statistics and reproducibility**. Statistical analysis was performed using GraphPad Prism 9 software. Data distribution was calculated using the *Shapiro-Wilk test*. Normally distributed data were presented as mean ± standard error (SEM) and analyzed by unpaired two-tailed parametric *t*-test. For non-normally distributed data, unpaired non-parametric Mann–Whitney *U* test was used, and as mean ± SEM were presented. For comparisons of more than two groups, two-way ANOVA test with multiple comparisons (normally distributed data) were used. Differences were considered statistically significant for $p < 0.05$. n refers to the number of biological replicates.

For qPCR, $n = 5$–10 was analyzed, data are representative of 2 experiments, *p* value was calculated with unpaired Student's *t* test

For deep RNA sequencing, $n = 3$ (two technical replicates) was analyzed, data are representative of 1 experiment. *P*-values were calculated with the Wald test and adjusted with the Benjamini–Hochberg analysis. Differential gene expression was considered significant with Benjamini–Hochberg adjusted *p*-value ≤ 0.05.

For WB, $n = 5$–15 was analyzed, data are representative of 15 experiments, *p* value was calculated with unpaired Student's *t* test or Mann–Whitney *U* test

For ELISA, $n = 6$ was analyzed, data are representative of 1 experiment, *p* value was calculated with unpaired Student's *t* test.

For BrdU assay, $n = 4$ was analyzed, data are representative of 1 experiment, *p* value was calculated with unpaired Student's *t* test.

For flow cytometry, $n = 5$–14 was analyzed, data are representative of 15 experiments, *p* value was calculated with unpaired Student's *t* test or Mann–Whitney *U* test.

For Caspase Glo 3/7 assay, $n = 4$ was analyzed, data are representative of 1–2 experiments, *p* value was calculated with unpaired Student's *t* test.

For Senescence Associated-β-Galactosidase assay, $n = 4$–6 was analyzed, data are representative of 2 experiments, *p* value was calculated with unpaired Student's *t* test.

For contraction assay, $n = 3$ was analyzed, data are representative of 4 experiments, *p* value was calculated with two-way ANOVA test with Sidak's multiple comparisons test.

For IHC, $n = 5$–24 was analyzed, data are representative of 14 experiments, *p* value was calculated with unpaired Student's *t* test or Mann–Whitney *U* test.

For echocardiography, $n = 18$–22 was analyzed, data are representative of 3 experiments, *p* value was calculated with unpaired Student's *t* test.

For radiotelemetry, $n = 10$–12 was analyzed, data are representative of 3 experiments, *p* value was calculated with unpaired Student's *t* test.

For ECG, $n = 11$ was analyzed, data are representative of 2 experiments, *p* value was calculated with unpaired Student's *t* test.

For Langendorff assay, $n = 4$ was analyzed, data are representative of 2 experiments, *p* value was calculated with paired *t*-test.

**Reporting summary**. Further information on research design is available in the Nature Portfolio Reporting Summary linked to this article.

## Data availability

RNA sequencing data were submitted to the ArrayExpress repository (accession code E-MTAB-12271). We provided all uncropped WB in Figure S13. The raw data for all graphs and charts have been uploaded in the https://zenodo.org/ (https://doi.org/10.7777/foo.bar, file name: Supplementary Data 1).

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

## Acknowledgements

We thank Mrs. Maria Comazzi-Fornallaz, Dr. Magdalena Diaz Ovalle, Dr. Zhongning Guo, and Dr. Andrea Laimbacher for excellent technical assistance, Sophistolab (Switzerland) and Genewiz (Germany) for excellent service support. Funding sources: G.K. acknowledges support from the Swiss National Science Foundation (310030_152876/1; 310030_175663; 310030_207708), Hartmann Muller Foundation, Theodor and Ida Herzog-Egli Foundation, Novartis Foundation, Swiss Life Foundation, Kurt and Senta Hermann Foundation, Hermann Klaus Foundation, and Stiftung für wissenschaftliche Forschung an der Universität Zürich.

## Author contributions

substantial contribution to the conception or design of the work: S.M., D.M., D.O., B.P., and K.G. the acquisition, analysis, or interpretation of data: S.M., D.M., R.M., O.C., R.F., P.E., G.F., S.P., U.S., O.E., R.D., K.K., H.J., D.O., B.P., and K.G. drafting the work: M.S., D.M., B.P., and K.G. substantial contribution in revision: H.A., D.M., B.P., and K.G.

## Competing interests

The disclosure statement of Prof. O. Distler (2 years backwards, 2020-2022): Prof. O. Distler has/had consultancy relationship with and/or has received research funding from or has served as a speaker for the following companies in the area of potential treatments for systemic sclerosis and its complications in the last two years: 4P-Pharma, Abbvie, Acceleron, Alcimed, Altavant, Amgen, AnaMar, Arxx, AstraZeneca, Blade, Bayer, Boehringer Ingelheim, Corbus, CSL Behring, Galderma, Galapagos, Glenmark, Gossamer, iQvia, Kymera, Lupin, Medscape, Merck, Miltenyi Biotec, Mitsubishi Tanabe; Novartis, Prometheus, Redxpharna, Roivant, and Topadur in the area of potential treatments of scleroderma and its complications. Patent issued "mir-29 for the treatment of systemic sclerosis" (US8247389, EP2331143). Research Grants: Kymera, Mitsubishi. All other authors declare no competing interests.
