## [Peer Review File · Communications Biology]

Reviewers' comments:

Reviewer #1 (Remarks to the Author):

In the current manuscript, Mara S. et al. reported Fos-related antigen 2 (Fosl-2) overexpression-driven cardiac fibrosis and arrhythmia development in Fosl-2 overexpressing (Fosl-2 Tg) mice compared to wild-type (Wt) mice with aging. Authors noted increased heart rate and the presence of arrhythmic indices in Fosl-2 Tg mice with impaired response under stress tests (Figure 2 and Figure 3A-3D). To dissect a molecular mechanism, authors showed reduced expression of mRNA transcripts in cardiac conduction system involved proteins (Figure 3E), altered localization of connexin 43 and E-cadherin at the level of intercalated discs (Figure 4C) with increased fibrosis in Purkinje fibers (Figure 4B) in 16-22 weeks old Fosl-2 Tg mice hearts. In the rest of the manuscript, the authors put substantial emphasis on their newly identified Lin-gp38+ cardiac fibroblasts proliferation and localization, co-localization with α SMA, Col1 α 1, Periostin, ADAM12, Cx43, Gja5, contactin-2 proteins in Fosl-2 Tg mice hearts (Figure 1f-1G, Figure 4, Supplementary Figure 4). To establish the role of the autoimmune response in gp38+ fibroblasts activation and noted cardiac pathologies in Fosl-2 Tg mice, the authors next generated Rag2^{-/-}Fosl-2 Tg mice lacking mature B and T cells. Notably, the observed cardiac fibrosis and arrhythmic indices were absent in Rag2^{-/-}Fosl-2 Tg mice (Figure 5C-5E, Supplementary Figure 11). Further, RNA-sequencing showed altered gene expression with pro-fibrotic transcriptomics in Lin-gp38+Fosl-2 Tg fibroblasts compared to Rag2^{-/-}Lin-gp38+ fibroblasts (Figure 6A-6B). The authors confirm the presence of gp38+ fibroblasts and increased expression of Fosl-2 in human systemic sclerosis patient heart sections and involvement of Fosl-2 for α SMA expression in the fetal human cardiac fibroblast cell line. After a careful review of the current manuscript, the present reviewer put the following points to be addressed:

1. Authors demonstrated that gp38+ fibroblasts co-localizes to conventional fibroblasts marker proteins, i.e., α SMA, Col1 α 1, Periostin, and ADAM12 (Figure 1H). However, the underlying mechanism and rationale for increased localization of gp38+ fibroblasts to gap junctions at cardiac intercalated discs (Figure 4C and 4D) and in Purkinje fibers (Figure 4B) in Fosl-2 Tg mice hearts are not convincing from the manuscript text. What type of injury provokes gp38+ fibroblasts localization to intercalated discs and Purkinje fibers in Fosl-2 Tg mice hearts? Did the authors' assess cell death/apoptosis in Fosl-2 Tg mice hearts at 7 weeks and 16-22 weeks of age? In addition, are there any differences in mononuclear immune cells infiltrate localization in Fosl-2 Tg mice hearts at 7 weeks and 16-22 weeks of age? TUNEL staining and H&E staining can provide additional information regarding these concerns.
2. In Figure 4B, the extent of gp38+ cells' co-localization to contactin-2+ cells is required to be quantified in Fosl-2 Wt, and Fosl-2 Tg mice heart fluorescent images as images are not convincing.
3. One of the critical findings in Fosl-2 Tg mice observed cardiac pathology is the involvement of autoimmune response. To this point, please report Figure 5B flow cytometry results in terms of detected individual immune cell types percent populations in individual dot plots with statistical analysis from Fosl-2 Wt and Fosl-2 Tg mice hearts.
4. In Figure 6A-6B and Supplementary Figure 12, please mention how many biologically independent mice and technical replicates per mice have been used to generate RNA-sequencing data from isolated fibroblasts.
5. In Figure 6A-6B and Supplementary Figure 12, please mention at what age gp38+ fibroblasts were FACS-sorted from Fosl-2 Tg, Rag2^{-/-}Fosl-2 Wt, and Rag2^{-/-}Fosl-2 Tg mice hearts.
6. The RNA sequencing data (metadata spreadsheet, processed data files, raw data files) are required to be submitted to NCBI Gene Expression Omnibus (GEO) repository for reviewers and readers' review. This requires editorial oversight to ensure transparent data sharing with the scientific community.
7. In Figure 6H, there is figure mislabeling between Fosl-2 Wt and Fosl-2 Tg fibroblasts. Please correct.
8. For Figure 6H, please specify how long fibroblasts were cultured before α SMA and stress fiber staining? Please specify culture conditions as well.

Overall, the current study findings are interesting and noteworthy. However, the above points are encouraged to be addressed to improve the manuscript credentials.

Reviewer #2 (Remarks to the Author):

The study confirms and extends published findings on the development of cardiac fibrosis in the hearts of *Fosl2* overexpressing mice (Venalis et al Arthritis Rheumatol 2015) and on the role of lymphocytes in the observed phenotype (Renoux et al Cell Rep 2020). The new findings relate to the documentation of arrhythmic and conduction defects. The following concerns are raised regarding the quality of the data:

Major:

1. It is impossible to appreciate fibrosis in the Tg mice. In control mice, the endomysial collagen network is not visualized due to low sensitivity of the staining strategy used by the authors. Thus, the Tg mouse simply appears (in some of the images) to have a normal collagen network. Better quality images and more sensitive methods of detection are needed.

2. Some experiments produce contradictory findings. In figure 2A, Tg mice have increased HR. However, in the echocardiographic data, HR is lower in Tg mice. Please explain.

3. α -SMA is expressed predominantly by VSMCs. Thus, the use of myocardial α -SMA levels to document fibrosis is highly problematic.

4. *Fosl2* Tg mice exhibit systemic inflammation and fibrotic changes in several different organs. Does reduction of cardiac fibrosis in mice lacking B/T cells reflect abrogation of systemic autoimmunity, or myocardial effects?

5. In the abstract, there is no obvious link between Systemic Sclerosis and the study. Thus, the first sentence seems unrelated to the manuscript. Please provide a background sentence explaining the focus on *fosl2*. Please also explain the basis for the experiment examining the role of lymphocytes.

6. *Fosl2* Tg mice responded "shorter" to ISO. The meaning of the sentence is unclear. Please revise and clarify.

7. In the figure showing human samples, the control shows no collagen in the myocardium. Moreover, what is the structure shown in the representative Ssc patient? These are not comparable representative images.

9. Figures: error bars are needed.

Minor:

Supplemental figure 1B: please show better quality WB.

Abstract: "in rheumatic disease systemic sclerosis", systemic sclerosis is sufficient.

"Analyzed mice were of mixed gender". Please indicate how many were male and how many female.

Reviewer #3 (Remarks to the Author):

The major claim of the paper is that *Fosl-2* over expression is associated with development of cardiac fibrosis and conduction abnormalities, and that this findings could be reversed by genetically depletion of B and T cells.

Regarding the relevance to the field, this works provides information about B and T cells, which are known have pro-fibrotic properties, however, the authors show their importance in the development of abnormal conduction.

The claims and the results go in line of what would be expected in a model like the one presented. Some details must be fixed before publishing the work:

-Bar charts do not include error bars, please add them

-In figure 6, the pathways are hard to read, the dark red color does not have good contrast with the text, also the font size is too small and when converted to image some words are hard to read.

Responses to the Reviewer #1

We would like to thank Reviewer #1 for helpful comments allowing us to improve our manuscript. Changes in the manuscript are highlighted in yellow.

Reviewer #1 (Remarks to the Author):

In the current manuscript, Mara S. et al. reported Fos-related antigen 2 (Fosl-2) overexpression-driven cardiac fibrosis and arrhythmia development in Fosl-2 overexpressing (Fosl-2 Tg) mice compared to wild-type (Wt) mice with aging. Authors noted increased heart rate and the presence of arrhythmic indices in Fosl-2 Tg mice with impaired response under stress tests (Figure 2 and Figure 3A-3D). To dissect a molecular mechanism, authors showed reduced expression of mRNA transcripts in cardiac conduction system involved proteins (Figure 3E), altered localization of connexin 43 and E-cadherin at the level of intercalated discs (Figure 4C) with increased fibrosis in Purkinje fibers (Figure 4B) in 16-22 weeks old Fosl-2 Tg mice hearts. In the rest of the manuscript, the authors put substantial emphasis on their newly identified Lin-gp38+ cardiac fibroblasts proliferation and localization, co-localization with α SMA, Colla1, Periostin, ADAM12, Cx43, Gja5, contactin-2 proteins in Fosl-2 Tg mice hearts (Figure 1f-1g, Figure 4, Supplementary Figure 4). To establish the role of the autoimmune response in gp38+ fibroblasts activation and noted cardiac pathologies in Fosl-2 Tg mice, the authors next generated Rag2-/-Fosl-2 Tg mice lacking mature B and T cells. Notably, the observed cardiac fibrosis and arrhythmic indices were absent in Rag2-/-Fosl-2 Tg mice (Figure 5C-5E, Supplementary Figure 11). Further, RNA-sequencing showed altered gene expression with pro-fibrotic transcriptomics in Lin-gp38+Fosl-2 Tg fibroblasts compared to Rag2-/-Lin-gp38+ fibroblasts (Figure 6A-6B). The authors confirm the presence of gp38+ fibroblasts and increased expression of Fosl-2 in human systemic sclerosis patient heart sections and involvement of Fosl-2 for α SMA expression in the fetal human cardiac fibroblast cell line. After a careful review of the current manuscript, the present reviewer put the following points to be addressed:

1. Authors demonstrated that gp38+ fibroblasts co-localizes to conventional fibroblasts marker proteins, i.e., α SMA, Colla1, Periostin, and ADAM12 (Figure 1H). However, the underlying mechanism and rationale for increased localization of gp38+ fibroblasts to gap junctions at cardiac intercalated discs (Figure 4C and 4D) and in Purkinje fibers (Figure 4B) in Fosl-2 Tg mice hearts are not convincing from the manuscript text. What type of injury provokes gp38+ fibroblasts localization to intercalated discs and Purkinje fibers in Fosl-2 Tg mice hearts?

Fosl-2^{tg} mice develop interstitial cardiac fibrosis with the presence of expanding gp-38⁺ fibroblasts within the whole heart, including the neighbourhood with gap junctions at cardiac intercalated discs and contactin-2⁺ Purkinje fibres. Fibroblasts expansion between cardiomyocytes causes the observed defects of the conduction system due to a loss of cardiac architecture and disruption of proteins composing gap junctions such as connexins. It has been previously demonstrated that the patients with pulmonary arterial hypertension (PAH) develop frequent arrhythmias including atrial fibrillation and sudden cardiac death. The authors concluded that an increase in inflammatory cytokines and fibrosis in the right heart contributes to dysfunction and predispose to arrhythmias in PAH (1). Another group investigated the effects of heart failure (HF) on Purkinje fibers using the volume-followed by pressure-overload rabbit model of HF with development of ventricular hypertrophy, diminished fractional shortening and ejection fraction, increased left ventricular dimensions, prolonged QRS and corrected QT and arrhythmic events, enhanced inflammation and fibrosis in the heart. They clearly showed that severe volume- followed by pressure-overload causes quickly progressing HF that reasons in extensive remodelling of Purkinje fibers (2).

Moreover, the interstitial fibrosis with reticular pattern is considered to have the greatest arrhythmogenic impact due to anisotropic re-entry based on loss of side-to-side electrical connections between cardiomyocytes (3). We demonstrated the interstitial fibrosis in Fosl-2^{tg} hearts. Fibrotic lesions can mechanically and electrically separate surviving bundles of cardiomyocytes that can cause arrhythmia by creating areas of conduction block or reduce conductivity between cardiomyocytes (4). In agreement with this notion, we demonstrated that Purkinje cells in

Fosl-2^{tg} mice were surrounded by accumulated collagen and cardiac fibroblast that separated them from other cell types. Therefore, we suggest that in Fosl-2^{tg} mice, progressive cardiac fibrosis (with an occurrence of collagen deposition and expansion of gp-38⁺ fibroblasts) destructs physiological gap junction network and thereby triggers arrhythmic events.

Additionally, we showed recently that activated cardiac fibroblasts (myofibroblasts), but not quiescent cardiac fibroblasts, promoted cardiac contraction rate by a direct stimulation of β -adrenoreceptor signalling in a model of fibrotic cardiac microtissues (5). Here, we showed that Fosl-2^{tg} gp38⁺ cardiac fibroblasts represented activated myofibroblasts, which without any stimulation revealed higher levels of α SMA total protein, α SMA-positive stress fibers, increased secreted pro-collagen I and increased contraction capacity (Figure 6F-I). Furthermore, as presented in Figure 6B, gene ontology analysis of upregulated genes in Rag-2^{-/-}Fosl-2^{tg} (vs. Rag-2^{-/-}Fosl-2^{wt}) fibroblasts indicated activation of profibrotic and rhythm-related pathways (Wnt interactions, tight-junction interactions, Bmal1-Clock activation circadian expression). All these changes in the biology of Fosl-2^{tg} gp38⁺ cardiac fibroblasts may predispose them to trigger pro-arrhythmic responses in neighborhood cardiomyocytes.

Did the authors' assess cell death/apoptosis in Fosl-2 Tg mice hearts at 7 weeks and 16-22 weeks of age? In addition, are there any differences in mononuclear immune cells infiltrate localization in Fosl-2 Tg mice hearts at 7 weeks and 16-22 weeks of age? TUNEL staining and H&E staining can provide additional information regarding these concerns.

Thank you for these suggestions. We performed the TUNEL staining and H&E staining. The H&E staining have been included in the Supplementary Figures 2A, 3A and 11A. TUNEL staining is presented as an Additional Figure 1 for Reviewer 1. There are almost no apoptotic cells in the hearts from 7- and 16-week-old Fosl-2^{wt} and Fosl-2^{tg} mice.

Additional Figure 1 to Reviewer 1. Assessment of the apoptotic cells in the heart of 7- and 16-week-old Fosl-2^{wt} and Fosl-2^{tg} mice

Representative pictures of TUNEL (Terminal Deoxynucleotidyl Transferase (TdT)-Mediated dUTP Nick End Labeling) assay (with Bond Refine Red Detection Kit) of myocardial sections from 7-week-old Fosl-2^{wt} and Fosl-2^{tg} mice (A), and 16-week-old Fosl-2^{wt} and Fosl-2^{tg} mice (B). Since the myocardium showed almost no apoptotic cells, we used as a positive control a lung sample from IPF patient (C). Apoptotic cells are visualized in pink and indicated by arrows (A-C). Scale bar: 100 μm.

We did not observe any differences in mononuclear immune cells infiltrating the myocardium of 7-week-old Fosl-2^{wt} and Fosl-2^{tg} mice. Contrary, we observed the differences in some populations of mononuclear immune cells between 16-week-old Fosl-2^{wt} and Fosl-2^{tg} hearts. The additional flow cytometry analyses are included in Figure 5B and we discussed these results in point 3 below.

2. In Figure 4B, the extent of gp38⁺ cells' co-localization to contactin-2⁺ cells is required to be quantified in Fosl-2 Wt, and Fosl-2 Tg mice heart fluorescent images as images are not convincing.

Thank you for this suggestion. There are no cells co-expressing gp38 and contactin-2. This is not surprising, because gp38 is a marker of fibroblasts and contactin-2 is a marker of Purkinje (cardiomyocyte) cells. Instead, we observed that in Fosl-2^{tg} hearts contactin-2⁺ Purkinje cells were surrounded by the deposited collagen (Figure 4A) and by the gp38⁺ fibroblasts (Figure 4B). Fibroblasts expansion affected the Purkinje-cardiomyocyte junctions that influences the cell-to-cell signalling, tissue architecture and in turn cell contractility (6). Therefore, we think that also in Fosl-2^{tg} hearts expanding gp38⁺ fibroblasts and accumulation of extracellular matrix disrupts Purkinje-cardiomyocyte junctions and cardiomyocyte-cardiomyocyte junctions (as shown in Figure 4C-D). This can influence signalling propagation and results in alterations in conduction system, development of arrhythmias, and rhythm defects.

3. One of the critical findings in Fosl-2 Tg mice observed cardiac pathology is the involvement of autoimmune response. To this point, please report Figure 5B flow cytometry results in terms of detected individual immune cell types percent populations in individual dot plots with statistical analysis from Fosl-2 Wt and Fosl-2 Tg mice hearts.

Previously, we showed that Fosl-2 overexpression induced T cell-mediated systemic inflammation in Fosl-2^{tg} mice and Fosl-2 was implicated in Treg development in the thymus (7). In this manuscript, we showed that in the Fosl-2^{tg} hearts there is a modest inflammation, demonstrated by statistically significant higher percentages of infiltrating Ly6G^{int}/Ly6C^{int} monocytes and Lin⁻/CD11b⁺ monocytes, lower percentages of CD11b⁺/MHCII^{hi} dendritic cells (DCs) and CD45⁺/Lin⁻ cells, and unchanged percentages of CD11b⁺/MHCII^{hi} DCs, Ly6G^{hi} neutrophils, T cells and Ly6c^{hi} monocytes (Figure 5B; gating strategy in Supplementary Figure 10). Following the Reviewer's suggestion, we added the flow cytometry analyses of individual immune cell types in Figure 5B.

4. In Figure 6A-6B and Supplementary Figure 12, please mention how many biologically independent mice and technical replicates per mice have been used to generate RNA-sequencing data from isolated fibroblasts.

We added the following information to the Supplementary Methods and Material: we used 3 mice for each mouse strain, each in 2 technical replicates.

5. In Figure 6A-6B and Supplementary Figure 12, please mention at what age gp38⁺ fibroblasts were FACS-sorted from Fosl-2 Tg, Rag2^{-/-}Fosl-2 Wt, and Rag2^{-/-}Fosl-2 Tg mice hearts.

Thank you for this suggestion. Accordingly, we added the information in the figure legends. We used the gp38⁺ fibroblasts from 16-20-week-old mice.

6. The RNA sequencing data (metadata spreadsheet, processed data files, raw data files) are required to be submitted to NCBI Gene Expression Omnibus (GEO) repository for reviewers and readers' review. This requires editorial oversight to ensure transparent data sharing with the scientific community.

Thank you for this suggestion, we submitted the RNA sequencing data to the ArrayExpress repository under the accession code E-MTAB-12271.

7. In Figure 6H, there is figure mislabeling between Fosl-2 Wt and Fosl-2 Tg fibroblasts. Please correct.

We apologize for this mislabelling; this has been corrected.

8. For Figure 6H, please specify how long fibroblasts were cultured before α SMA and stress fiber staining? Please specify culture conditions as well.

We added this information in the Supplementary Material and Methods in subchapter *Immunofluorescent staining*. “For α SMA immunofluorescence staining combined with Phalloidin–Tetramethylrhodamine B isothiocyanate (Sigma) staining, cells were seeded in 8-chamber glass slides (Lab-Tec) at the density of 10000 cells/well. After 24 hours of culture, the cells were stimulated with or without TGF- β for 72 hours.”

Overall, the current study findings are interesting and noteworthy. However, the above points are encouraged to be addressed to improve the manuscript credentials.

We would like to thank Reviewer 1 for the positive feedback. We hope that we improved the manuscript quality and credentials.

Responses to the Reviewer #2

We would like to thank Reviewer #2 for helpful comments allowing us to improve our manuscript. Changes in the manuscript are highlighted in yellow.

Reviewer #2 (Remarks to the Author):

The study confirms and extends published findings on the development of cardiac fibrosis in the hearts of Fosl2 overexpressing mice (Venalis et al Arthritis Rheumatol 2015) and on the role of lymphocytes in the observed phenotype (Renoux et al Cell Rep 2020). The new findings relate to the documentation of arrhythmic and conduction defects. The following concerns are raised regarding the quality of the data: Major:

1. It is impossible to appreciate fibrosis in the Tg mice. In control mice, the endomysial collagen network is not visualized due to low sensitivity of the staining strategy used by the authors. Thus, the Tg mouse simply appears (in some of the images) to have a normal collagen network. Better quality images and more sensitive methods of detection are needed.

We apologize for the low-quality pictures. In the revised version we exchanged them with better quality pictures. In the low magnification pictures, the endomysial collagen network is poorly visible, therefore, we present new pictures with higher magnification. Additionally for the Reviewer 2 we present 1000x magnification to visualise collagen network.

Additional Figure 1 for Reviewer 2. Endomysial collagen network in the myocardial sections from 16-week-old Fosl-2^{wt} mouse.

2. Some experiments produce contradictory findings. In figure 2A, Tg mice have increased HR. However, in the echocardiographic data, HR is lower in Tg mice. Please explain.

We observed increased baseline HR in conscious freely moving Fosl-2^{tg} mice using radiotelemetry. Such increased HR in combination with the reduction in heart rate variability (SDNN, RMSSD, NN, pNN6, NN6) indicated a higher sympathetic tone. Furthermore, Fosl-2^{tg} mice and isolated ex-vivo Fosl-2^{tg} hearts showed impaired stress responses, further indicating defects in the regulation by the autonomic nervous system. Mice undergoing the echocardiography measurements were anesthetized that may cause a stressful condition to mice. Therefore, the lower HR observed in Fosl-2^{tg} mice under the anesthesia confirmed their aberrant response to stress.

3. α -SMA is expressed predominantly by VSMCs. Thus, the use of myocardial α -SMA levels to document fibrosis is highly problematic.

We agree with this comment. We are also aware that using WB or IHC we cannot precisely distinguish between α SMA-expressing VSMCs and α SMA-expressing fibroblasts. However, we think that the baseline α SMA expression in VSMCs and in fibroblasts should be similar in young mice without a gross phenotype. Indeed, as it is shown in Supplementary Figure 3B (IHC) and in Supplementary Figure 3C (WB), there is similar expression levels of α SMA in the hearts from 7-week-old Fosl-2^{wt} and Fosl-2^{tg} mice. Contrary, in Figure 1B we observed statistically significant increase in the amount of α SMA protein in the hearts from 16-week-old Fosl-2^{tg} mice compared to 16-week-old Fosl-2^{wt}. We may remove the data with α SMA expression measured in the whole heart if Reviewer 2 suggests this.

Additionally, in the Figure 1H, in Fosl-2^{tg} heart we showed the co-expression of gp-38-expressing cardiac fibroblasts not only with α -SMA but also with other myofibroblast markers as vimentin, ADAMS-12 and periostin. Contrary, in the Fosl-2^{wt} heart α -SMA expression was limited to VSMCs.

We evaluated mainly the level of fibrosis based on collagen (S. Red) and gp-38 staining. We used α -SMA as an additional marker of myofibroblasts.

Moreover, in Figure 6H we presented statistically significant higher protein levels of α SMA measured by WB in cardiac Fosl-2^{tg} fibroblasts compared to Fosl-2^{wt} fibroblasts. Similarly, in Figure 7H, we showed statistically significant higher protein levels of α SMA measured by WB in human cardiac Fosl-2-overexpressing fibroblasts compared to the control cells. These results suggest that α SMA protein levels upregulated in activated cardiac fibroblasts (myofibroblasts).

4. Fosl2 Tg mice exhibit systemic inflammation and fibrotic changes in several different organs. Does reduction of cardiac fibrosis in mice lacking B/T cells reflect abrogation of systemic autoimmunity, or myocardial effects?

Our study demonstrated that the autoimmune response was necessary to trigger cardiac fibrogenesis and conduction system alterations in Fosl-2^{tg} mice. Fosl-2 overexpression in the absence of systemic inflammation, as shown in Rag2^{-/-}Fosl-2^{tg} mice, neither promoted cardiac fibrosis nor alteration in the conduction system and gap junction architecture. Similarly, the RNAseq analysis confirmed that inflammation strongly affects cardiac fibroblasts by not only enhancing fibrotic changes, but also activating pathways implicated in arrhythmogenic alterations. Our results are consistent with several other studies, where mice with defects in adaptive immunity developed cardiac hypertrophy but were protected from ventricular dilation and adverse cardiac remodelling (8).

5. In the abstract, there is no obvious link between Systemic Sclerosis and the study. Thus, the first sentence seems unrelated to the manuscript. Please provide a background sentence explaining the focus on fosl2. Please also explain the basis for the experiment examining the role of lymphocytes.

Thank you for this comment. Accordingly, we made the changes in the Abstract, and we hope that the text is written in clearer way in the corrected version, and that Reviewer 2 may accept these changes.

6.Fosl2 Tg mice responded “shorter” to ISO. The meaning of the sentence is unclear. Please revise and clarify.

We apologize for this unfortunate sentence, we corrected it in the manuscript as following:

“Moreover, Fosl-2^{tg} mice showed reduced response duration to ISO stimulation (Fig.3B-C, Suppl.Fig.8A).”

7.In the figure showing human samples, the control shows no collagen in the myocardium. Moreover, what is the structure shown in the representative Ssc patient? These are not comparable representative images.

There are small endocardial biopsies from the left ventricle, for the controls the biopsies originated from the patients with healed myocarditis. According to the Reviewer 2's suggestion we performed another S. Red staining on more representative myocardial biopsy for the healed myocardium.

8.Figures: error bars are needed.

Thank you for this suggestion, we added error bars to each graph.

Minor:

Supplemental figure 1B: please show better quality WB.

We replaced the WB with a better-quality picture.

Abstract: “in rheumatic disease systemic sclerosis”, systemic sclerosis is sufficient.

We corrected this accordingly.

“Analyzed mice were of mixed gender”. Please indicate how many were male and how many female.

In the “Methods: *Murine and human specimens*” we specified that we used 60% of males and 40% of females.

Responses to the Reviewer #3

We would like to thank Reviewer #3 for helpful comments allowing us to improve our manuscript. Changes in the manuscript are highlighted in yellow.

Reviewer #3 (Remarks to the Author):

The major claim of the paper is that Fos1-2 over expression is associated with development of cardiac fibrosis and conduction abnormalities, and that this findings could be reversed by genetically depletion of B and T cells. Regarding the relevance to the field, this works provides information about B and T cells, which are known have pro-fibrotic properties, however, the authors show their importance in the development of abnormal conduction. The claims and the results go in line of what would be expected in a model like the one presented.

Thank you very much for the positive feedback.

Some details must be fixed before publishing the work:

-Bar charts do not include error bars, please add them

Thank you for this suggestion, accordingly we included error bars to each graph.

-In figure 6, the pathways are hard to read, the dark red color does not have good contrast with the text, also the font size is to small and when converted to image some words are hard to read.

We changed these graphs, we hope that the quality is now better and the graphs are easier to read.

References:

1. P. S. Salva, G. E. Bacon, Parents and steroid use by nonathletes. *Pediatrics* **84**, 940-941 (1989).
2. S. Logantha *et al.*, Remodeling of the Purkinje Network in Congestive Heart Failure in the Rabbit. *Circ Heart Fail* **14**, e007505 (2021).
3. M. Stein *et al.*, Dominant arrhythmia vulnerability of the right ventricle in senescent mice. *Heart Rhythm* **5**, 438-448 (2008).
4. W. G. Stevenson, Ventricular scars and ventricular tachycardia. *Trans Am Clin Climatol Assoc* **120**, 403-412 (2009).
5. P. Blyszczuk *et al.*, Activated Cardiac Fibroblasts Control Contraction of Human Fibrotic Cardiac Microtissues by a beta-Adrenoreceptor-Dependent Mechanism. *Cells* **9**, (2020).
6. V. Garcia-Bustos *et al.*, Changes in the spatial distribution of the Purkinje network after acute myocardial infarction in the pig. *PLoS One* **14**, e0212096 (2019).
7. F. Renoux *et al.*, The AP1 Transcription Factor Fos12 Promotes Systemic Autoimmunity and Inflammation by Repressing Treg Development. *Cell Rep* **31**, 107826 (2020).
8. F. Laroumanie *et al.*, CD4⁺ T cells promote the transition from hypertrophy to heart failure during chronic pressure overload. *Circulation* **129**, 2111-2124 (2014).

REVIEWERS' COMMENTS:

Reviewer #1 (Remarks to the Author):

In their revised manuscript, the authors substantially and satisfactorily addressed my previous review points. Overall, this manuscript is well-written and provides novel and substantial evidence through complimentary experimentation results regarding the role of Fos1-2 in cardiac physiology and attains credentials to be considered for immediate publication.

Reviewer #2 (Remarks to the Author):

The authors have addressed my concerns. I have no further recommendations.